# LSD1 inhibition circumvents glucocorticoid-induced muscle wasting of male mice

Qingshuang Cai [1], Rajesh Sahu [1], Vanessa Ueberschlag-Pitiot[1], Sirine Souali-Crespo [1], Céline Charvet [1], Ilyes Silem[1], Félicie Cottard [1], Tao Ye [1], Fatima Taleb[1], Eric Metzger [2], Roland Schuele [2], Isabelle M. L. Billas [1], Gilles Laverny [1], Daniel Metzger [1] & Delphine Duteil [1] ✉

Synthetic glucocorticoids (GC), such as dexamethasone, are extensively used to treat chronic inflammation and autoimmune disorders. However, long-term treatments are limited by various side effects, including muscle atrophy. GC activities are mediated by the glucocorticoid receptor (GR), that regulates target gene expression in various tissues in association with cell-specific co-regulators. Here we show that GR and the lysine-specific demethylase 1 (LSD1) interact in myofibers of male mice, and that LSD1 connects GR-bound enhancers with NRF1-associated promoters to stimulate target gene expression. In addition, we unravel that LSD1 demethylase activity is required for triggering starvation- and dexamethasone-induced skeletal muscle proteolysis in collaboration with GR. Importantly, inhibition of LSD1 circumvents muscle wasting induced by pharmacological levels of dexamethasone, without affecting their anti-inflammatory activities. Thus, our findings provide mechanistic insights into the muscle-specific GC activities, and highlight the therapeutic potential of targeting GR co-regulators to limit corticotherapy-induced side effects.

Since the first clinical use in the late 1940s, glucocorticoids (GC) have revolutionized the field of medicine. Indeed, synthetic GC, such as prednisone and dexamethasone (DEX), are prescribed for many chronic inflammatory conditions[1], including colitis, rheumatoid arthritis, multiple sclerosis and asthma, as well as for immunosuppression in organ-transplanted patients[2]. It is estimated that 1–3% of the population of Western countries are long-term users of GC[3–6]. However, systemic treatments are limited by various side effects, including diabetes, osteoporosis and muscle wasting[7]. As to date, none of the many synthetic GC analogues has anti-inflammatory effects that are fully dissociated from iatrogenic effects, new therapeutic strategies are in demand.

GC activities are mediated by the glucocorticoid receptor (GR; NR3C1), a member of the nuclear receptor superfamily. GC induce the translocation of GR into the nucleus, where it binds to its response elements (GREs and nGREs) and recruits various co-factors[8–10], or interferes with other transcription factor activity, to regulate target gene expression[11,12]. Interestingly, recent studies indicate that various transcriptional co-regulators play a key role in GR cell-specific effects. For instance, GR has been shown to interact with FOXA2 in hepatocytes to promote gluconeogenesis[13], with FOXA1 in prostate cancer cells to stimulate androgen receptor (AR)-regulated pathways under androgen-deprived conditions[14], with MYOD1 in muscle fibers to negatively regulate muscle mass[15], and with forkhead box O1 (FOXO1) to induce muscle atrophy in C2C12 cells exposed to DEX[16].

Lysine-specific demethylase 1 (LSD1, also called KDM1A) is a flavin adenine dinucleotide (FAD)-dependent amine oxidase involved in transcriptional gene regulation[17,18]. Within the repressive CoREST complex, LSD1 demethylates mono- and di-methylated histone H3 at lysine 4 (H3K4me1 and H3K4me2) to inhibit transcription, whereas it

[1]Université de Strasbourg, CNRS, Inserm, IGBMC UMR 7104- UMR-S 1258, F-67400 Illkirch, France. [2]Klinik für Urologie und Zentrale Klinische Forschung, Klinikum der Albert-Ludwigs-Universität Freiburg, D-79106 Freiburg, Germany. ✉e-mail: duteild@igbmc.fr

promotes gene expression via demethylation of mono- and di-methylated histone H3 at lysine 9 (H3K9me1 and H3K9me2) in collaboration with various transcription factors[19,20]. A recent study revealed that LSD1 is part of the GR nuclear complex in the presence of DEX and demethylates H3K4me2 in the lung adenocarcinoma cell line A549[21]. A direct interaction between LSD1 and GR was also reported in HeLa cells[21], and LSD1 was shown to be essential for muscle fiber regeneration upon injury[22]. Since LSD1 controls the activity of various transcription factors, mediates GC-dependent metabolic reprogramming during myogenic differentiation[23], and can be targeted with inhibitors[24–26], this co-factor is a potential candidate to modulate GR activity in a cell-specific manner.

Here we show that LSD1 interacts with GR at enhancer regions in skeletal muscles at both physiological and pharmacological GC levels, and establishes a functional link with the transcription factor NRF1 at promoter sites to modulate the expression of genes controlling anti-anabolic and catabolic pathways. The interplay between LSD1 and GR in myofibers is also crucial for promoting muscle atrophy during the first hours following nutrient deprivation. Importantly, our findings demonstrate that the selective LSD1 inhibitor CC-90011 attenuates DEX-induced muscular atrophy, without affecting its anti-inflammatory properties. Consequently, targeting GR co-activators such as LSD1 opens therapeutic opportunities to circumvent deleterious consequences associated with GC administration.

## Results

### LSD1 interacts with GR to control target gene expression in mouse myofibers at physiological GC levels

To unveil whether LSD1 regulates GR activity in skeletal muscles under physiological conditions, we characterized GR and LSD1 expression, and found that both proteins were present at similar levels in muscles composed mainly of slow oxidative fibers (e.g., soleus) or fast-glycolytic fibers (e.g., gastrocnemius, tibialis and quadriceps) (Supplementary Fig. 1a). GR and LSD1 were mainly located in the skeletal muscle nuclei of both slow and fast-twitch myofibers, in which they colocalize (Fig. 1a and Supplementary Fig. 1b) and interact as revealed by co-immunoprecipitation experiments (Fig. 1b). To determine GR and LSD1 genomic localization in limb muscles, we conducted ChIP-seq experiments on 9-week-old mice using anti-LSD1 and anti-GR antibodies. Peak calling analysis revealed a total of 16,616 LSD1-binding sites (LSDBS) distributed across 9057 genes, and 14,108 GR-binding sites (GRBS) located in 7486 genes. LSDBS were evenly distributed across the genome [24% in intergenic, 35% in intronic and 33% in proximal promoter regions (TSS, −1 kb to +100 bp)], a repartition similar to that of GRBS in skeletal muscles (Supplementary Fig. 1c), in agreement with our previous studies[15]. Importantly, bedtools analysis revealed that more than half of GR peaks share the same locations with LSD1 peaks. In addition, 5649 genes were bound by both proteins (Supplementary Fig. 1d), and SeqMINER analysis revealed that about 80% of the DNA segments bound by LSD1 correlate with GR occurrence (Fig. 1c, d). Pathway analysis revealed that these common genes belong to insulin and EGFR1 signaling, and nuclear receptor signaling among other networks (Supplementary Fig. 1e). Examples of genomic regions of previously described GC target genes[16,27] bound by both GR and LSD1 (e.g., Ddit4, Trim63 and Fbxo32, also known as Redd1, Murf1 and Atrogin1, respectively) are depicted in Fig. 1e and Supplementary Fig. 1f. In the Ddit4 locus, we identified four binding sites shared by GR and LSD1, three located at 17−27 kb upstream of the TSS, referred to as GBSe1, GBSe2 and GBSe3, and one at the promoter region (GBSp1) (Fig. 1e), and ChIP-qPCR analysis confirmed the presence of LSD1 and GR at these sites (Supplementary Fig. 1g). Whereas GBSe2 (5′-AGAACAttgTGTTCT-3′) corresponds to a consensus GRE[15], GBSe1 (5′-GGAACAgcaTGTGCA-3′) and GBSe3 (5′-AGAACGctcTGTACC-3′) differ from the consensus on the second and the two half-sites, respectively, and no GRE-like sequence could be identified for GBSp1 that mainly

encompasses GC-repeats. To demonstrate that LSD1 and GR co-localize on the Ddit4 GBSs, we performed a two-step ChIP (Re-ChIP) experiment in mouse skeletal muscles. This analysis revealed an enrichment at the four GBSs when skeletal muscle nuclear extracts were first immunoprecipitated with an anti-GR antibody and then with an anti-LSD1 antibody, or vice versa (Fig. 1f). Altogether, these results show that GR and LSD1 interact at genomic loci of GC target genes in skeletal muscles.

To characterize LSD1 function in skeletal muscles, we generated LSD1[skm-/-] mice, by intercrossing mice bearing floxed LSD1 first exon with the myofiber-specific HSA-Cre deleter strain (Supplementary Fig. 1h). Western blot and immunofluorescence analyses revealed that LSD1 was efficiently ablated in myofibers of LSD1[skm-/-] mice (Supplementary Fig. 1i, j). Notably, hematoxylin and eosin (H&E) staining showed that the histology of 9-week-old LSD1[skm-/-] muscles was similar to that of control mice (Supplementary Fig. 1k). Furthermore, body, muscle and adipose tissue weights, as well as muscle strength, were not significantly affected at 9 weeks by LSD1 loss in myofibers (Supplementary Fig. 1l, m). However, RNA sequencing (RNA-seq) on gastrocnemius muscles from 9-week-old control and LSD1[skm-/-] mice identified 2669 differentially expressed genes (reads > 50 and p < 0.05). Bioinformatic analyses combining ChIP-seq and RNA-seq datasets showed that LSD1 is bound to 742 genes that are up-regulated in skeletal muscles of LSD1[skm-/-] mice, and 845 that are down-regulated, that are associated with various pathways including "Cushing syndrome" and "Focal adhesion", and "Mitophagy" and "Autophagy", respectively (Supplementary Fig. 1n). In addition, the overlap between genes bound by both GR and LSD1 in skeletal muscles, and genes downregulated upon LSD1 ablation, revealed that GR and LSD1 are co-recruited to half of the LSD1 target genes (Supplementary Fig. 1o). Moreover, these results combined with our previously published RNA-seq data of myofiber GR-deficient mice (hereafter GR[(i)skm-/-] mice)[15] revealed that half of the genes down-regulated in gastrocnemius of GR[(i)skm-/-] mice are bound by both GR and LSD1 (Supplementary Fig. 1p). In addition, the comparison between GR and LSD1 transcriptomes unveiled that 778 genes are down-regulated and 456 are up-regulated in skeletal muscles of both LSD1[skm-/-] mice and GR[(i)skm-/-] mice (Fig. 1g). Pathway analysis of the genes down-regulated in both mutant mice uncovered FOXO signaling (e.g., Foxo4, Gsk3b, Sirt1, Stat3 and Ubc), autophagy (e.g., Becn1, Bnip3, Ctse and Ulk2), mitophagy (e.g., Cited2, Mfn1, Nbr1 and Usp15) and insulin signaling pathways (e.g., Calm1, Pik3r1, Pik3r3 and Tsc22d1) (Fig. 1h), whereas the genes upregulated in both mutant mice belonged to pathways including focal adhesion (e.g., Chad, Col1a1, Lama4 and Mylk), hepatocyte growth factor signaling (e.g., Hras, Itga1, Pak1 and Rasa1) and PI3K-Akt-mTOR pathway (e.g., Fgf1, Fgf11, Mapk3 and Rps6) (Fig. 1i). Altogether, our data show that GR and LSD1 interact at a large number of loci in myofibers at physiological GC levels to control both anabolic and catabolic pathways.

### LSD1 bridges GR at active enhancers and NRF1 at active promoters to control target gene expression

The seqMINER-generated heatmap of LSD1, GR and our previously published histone mark datasets[15,28] showed that about 42% of the LSDBS were located at active promoters (7044 peaks), defined by the presence of H3K4me2 and H3K4me3, and low H3K4me1 levels, and 42% at active enhancers (7026 peaks), characterized by the presence of H3K4me1 and H3K4me2, and low H3K4me3 levels (Fig. 2a). A similar analysis revealed that more than half of GRBS were located at active enhancers, and 31% of them at active promoters (Supplementary Fig. 2a), in accordance with our previously reported datasets[15]. Hypergeometric optimization of motif enrichment (HOMER) de novo analysis of LSDBS revealed that LSD1-associated enhancer regions encompass binding sites for the transcription factors MEF2C, SIX2 and GR (Fig. 2b). In agreement with our previous work[15], GR was bound to GREs at enhancers (Supplementary Fig. 2b). In contrast, most of the GR and

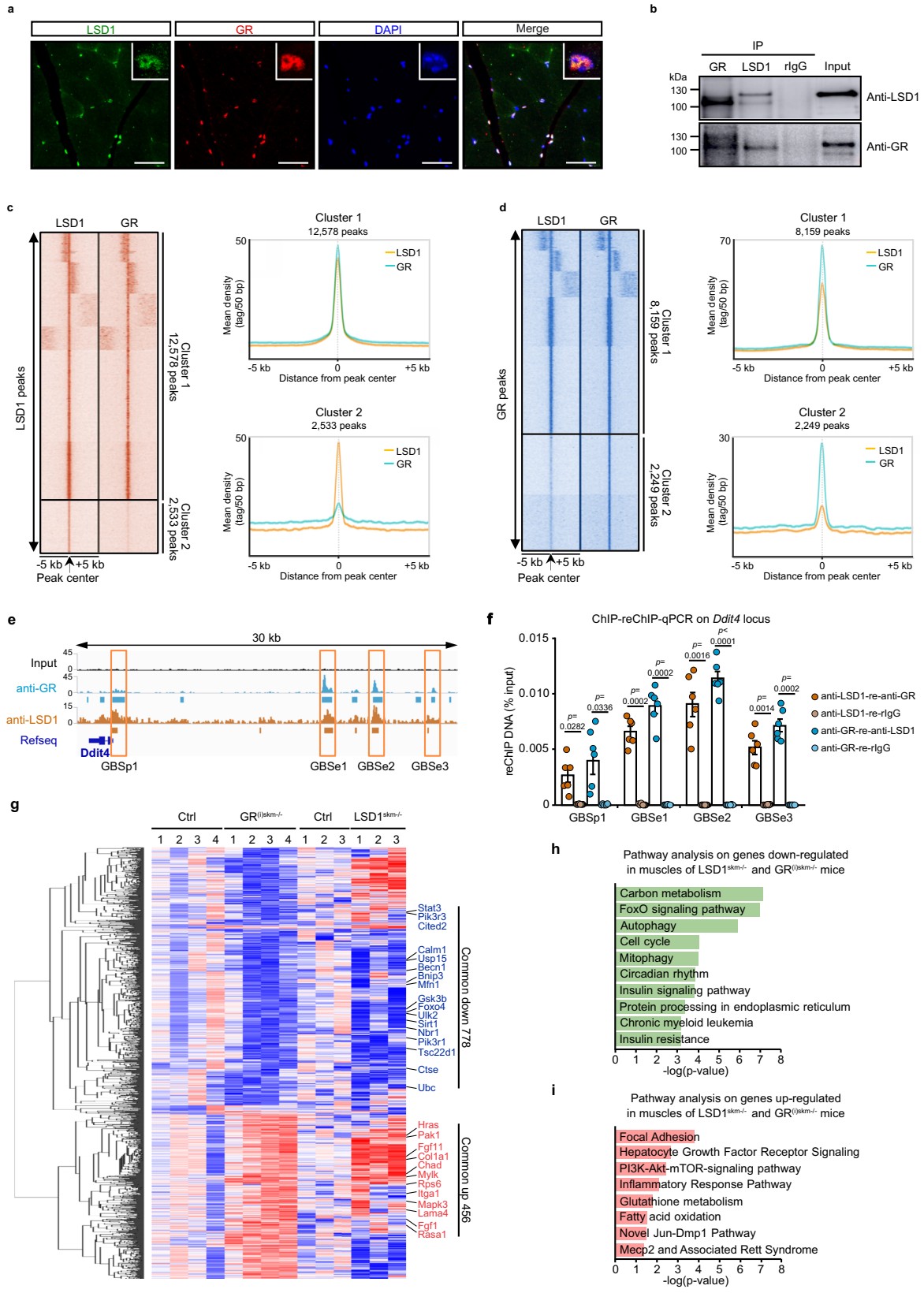

LSD1 co-occupied promoter regions were GC-rich elements, including the motif for the transcription factor nuclear respiratory factor 1 (NRF1, 5′-GCGCatGCGC-3′) (Fig. 2c and Supplementary Fig. 2c). Co-immuno-precipitation experiments revealed that LSD1 was immunoprecipitated by anti-NRF1 antibodies in mouse skeletal muscle nuclear extracts

(Fig. 2d), and immunoprecipitation of purified recombinant LSD1 in the presence of purified recombinant NRF1 protein revealed that the interaction between both proteins is direct (Fig. 2e).

Importantly, whereas NRF1 was immunoprecipitated by GR anti-bodies in muscle nuclear extracts, in agreement with our previous

**Fig. 1 | LSD1 interacts with GR to control target gene expression in mouse myofibers at physiological glucocorticoid levels. a** Representative immunofluorescent detection of LSD1 (green) and GR (red) in gastrocnemius muscles of 9-week-old wild-type mice. Nuclei were stained with DAPI. A zoomed-in view of the confocal observation is shown on the top right panel. Scale bar, 50 μm. $N = 3$. **b** Representative western blot analysis of GR and LSD1 co-immunoprecipitation in gastrocnemius muscle nuclear extracts. Rabbit IgG served as a control for immunoprecipitation. $N = 3$ mice. Note that the observed discrepancies in the molecular weight on membranes decorated with anti-LSD1 antibody originate from the high sensitivity of LSD1 to salt and/or pH composition of the elution buffer. **c, d** Tag density map of LSD1 and GR in skeletal muscles, +/− 5 kb from the LSD1 (**c**) or the GR (**d**) peak center, and corresponding average tag density profiles. **e** Localization of

GR and LSD1 at the *Ddit4* locus. The four GR binding sites located at enhancer (GBSe1, GBSe2 and GBSe3) and promoter (GBSp1) regions are boxed in orange. **f** Two-step chromatin immunoprecipitation performed with indicated antibodies followed by qPCR analysis (ChIP-reChIP-qPCR) in gastrocnemius muscles of wild-type mice at GBSe1, GBSe2, GBSe3 and GBSp1 of *Ddit4*. $N = 6$ biological replicates. Mean ± SEM. Two-way ANOVA with Tukey correction. **g** Heatmap with hierarchical clustering depicting the mean centered normalized expression of genes differentially expressed in RNA-seq in gastrocnemius muscles of 9-week-old Ctrl, GR[(i)skm-/-] and LSD1[skm-/-] mice. **h, i** Pathway analysis of genes down- (**h**) and up-regulated (**i**) in gastrocnemius muscles of both GR[(i)skm-/-] and LSD1[skm-/-] mice at 9 weeks, with *p* values adjusted using the Benjamini-Hochberg. Source data are provided as a Source Data file.

results[15], NRF1 and GR did not interact in muscles of LSD1[skm-/-] mice (Fig. 2f), indicating that LSD1 is required for this interaction. In agreement with these results, ChIP-qPCR experiments revealed that both GR and NRF1 are recruited to GBSe2 of the *Ddit4* gene in skeletal muscles of control mice, and that NRF1 recruitment is decreased by more than 70% in LSD1[skm-/-] mice, whereas that of GR was not affected (Fig. 2g). Moreover, whereas GR and NRF1 were detected at the *Ddit4* GBSp1 in controls, GR binding was impaired in LSD1-depleted myofibers, whereas that of NRF1 was not (Fig. 2g). Of note, as a negative control, we found that GR and NRF1 were not enriched at the promoter region of the *Pax7* gene that was not identified as a GR or LSD1 target by ChIP-seq (Fig. 2g and Supplementary Fig. 2d).

To determine if the DNA segments bound GR and LSD1 are bound by NRF1 as well, we conducted a ChIP-seq analysis on skeletal muscle nuclear extracts, identifying 3130 NRF1 binding sites (Supplementary Fig. 2e). HOMER motif analysis confirmed the predominance of the NRF1 motif at NRF1-bound genomic locations (Fig. 2h). Most of the NRF1 peaks were located at promoter regions (−1000 bp to 100 bp from TSS), corresponding to 2955 genes. These genes were mainly associated with pathways involved in ubiquitin-mediated proteolysis and OXPHOS (Supplementary Fig. 2e, f). Our studies revealed that the presence of GR and LSD1 at promoters strongly correlates with that of NRF1, in contrast to GR-LSD1 bound enhancers for which NRF1 binding was not detected, as exemplified for various catabolic genes (Fig. 2i, j and Supplementary Fig. 2g). These findings were further supported by ChIP-qPCR experiments conducted at multiple loci (Fig. 2g and Supplementary Fig. 2h). Analysis of the overlaps between GR, LSD1 and NRF1 target genes unveiled their collective regulation of more than 1000 genes, mainly involved in the protein degradation (Fig. 2k, l). To further characterize the interplay between GR, LSD1 and NRF1 on gene regulation, we performed small interfering RNA (siRNA)-mediated knock-down of these factors in C2C12 myotubes. Notably, while reduced NRF1 expression did not alter GR and LSD1 levels (Supplementary Fig. 2i), silencing of any of these factors led to approximately a 50% decrease in the expression of the GR targets *Ddit4* and *Fbxo31* (Fig. 2m). Together, these results show that LSD1 is instrumental for the interaction between GR at enhancers and NRF1 at promoter regions to stimulate target gene expression in skeletal muscles at physiological GC levels.

## The GR/LSD1 complex promotes starvation-induced muscle atrophy

To determine whether LSD1 controls GR transcriptional activity at high endogenous GC levels, we performed a starvation challenge that activates the hypothalamic-pituitary-adrenal (HPA) axis and stimulates the production of GC by the adrenal gland. As shown in Fig. 3a, LSD1 and GR interaction is maintained in gastrocnemius muscles after 12 and 24 h of fasting. To characterize LSD1 function in skeletal muscles during food deprivation, we generated LSD1[(i)skm-/-] mice, in which LSD1 was selectively ablated in myofibers at adult stage using the tamoxifen-dependent CreER[T2] system[29] (Supplementary Fig. 3a–c) to avoid potential developmental effect of the mutation. After 48 h of fasting, a

similar reduction in body and spleen mass was observed in control, GR[(i)skm-/-] and LSD1[(i)skm-/-] mice, while a more pronounced decrease in white adipose tissue (WAT) mass was noted in the mutant groups relative to the control ones (Fig. 3b and Supplementary Fig. 3d). H&E staining of epididymal WAT revealed a reduced lipid droplet size in GR[(i)skm-/-] and LSD1[(i)skm-/-] mice compared to controls (Supplementary Fig. 3e). While after a 48 h starvation control mice exhibited decreased muscle strength accompanied by a significant reduction of the mass of gastrocnemius, tibialis and quadriceps fast-switch muscles, and of gastrocnemius cross-sectional area (CSA), these differences were not observed in GR and LSD1 mutant mice (Fig. 3b–e and Supplementary Fig. 3d, f–h). Of note, the mass of slow-twitch soleus muscles was unaffected by starvation (Fig. 3b and Supplementary Fig. 3d), in agreement with former studies[27,30,31].

Next, we assessed the impact of GR and LSD1 loss on anabolic and catabolic pathways upon food deprivation. Phosphorylation of MTOR and its downstream effector, EIF4EBP1 (hereafter named 4E-BP1), was decreased in starved control mice. In contrast, MTOR and 4E-BP1 phosphorylation levels were not affected by food deprivation in GR[(i)skm-/-] and LSD1[(i)skm-/-] mice, suggesting that protein synthesis is not reduced in mutant mice (Fig. 3f, g and Supplementary Fig. 3i–k). In addition, transcript levels of the TORC1 negative regulator *Ddit4* were strongly increased in starved control mice, but not in mutants (Fig. 3h and Supplementary Fig. 3l), thus suggesting that GR and LSD1 decrease TORC1 activity through DDIT4 in myofibers. To further investigate the molecular pathways controlling starvation-induced muscular atrophy, transcript levels of genes involved in protein degradation were analyzed by RT-qPCR in gastrocnemius muscles. *Myostatin* expression, which is stimulated during muscle atrophy[32–35], was more than 3-fold induced by food deprivation in control mice, but not in mutants, thus demonstrating that starvation-induced *myostatin* expression is myofiber GR and LSD1-dependent (Fig. 3h and Supplementary Fig. 3l). In addition, FOXO3A and GSK3B, which stimulate catabolism, were activated by dephosphorylation in fasted control mice. In contrast, these two factors were mainly found in their inactive form in GR[(i)skm-/-] and LSD1[(i)skm-/-] mice (Fig. 3f, g and Supplementary Fig. 3i–k). In agreement, transcript levels of *Ubiquitin C (Ubc)*, *Atrogin1 (Fbxo32)* and *Murf1 (Trim63)* from the ubiquitin-proteasome system (UPS) were strongly induced by food deprivation in control mice, contrary to mice lacking GR or LSD1 in myofibers (Fig. 3h and Supplementary Fig. 3l), showing that GR and LSD1 in myofibers are required for the induction of genes involved in protein degradation. Moreover, even though LC3I/II and p62 protein levels were similar among genotypes, the expression of *Bnip3*, *Cathepsin L (Cstl)* and *Gabaralp1* from the autophagy pathway was higher in starved control mice, but not induced in GR[(i)skm-/-] and LSD1[(i)skm-/-] mice (Fig. 3f, g and Supplementary Fig. 3i–k). Together, these data show that GR and LSD1 in myofibers control a similar transcriptional repertoire involved in food deprivation-induced impairment of protein synthesis and the induction of protein degradation. In agreement, ultrastructural analysis of gastrocnemius muscles revealed disruptions of myofibrils in more than 60% of the sarcomeres after

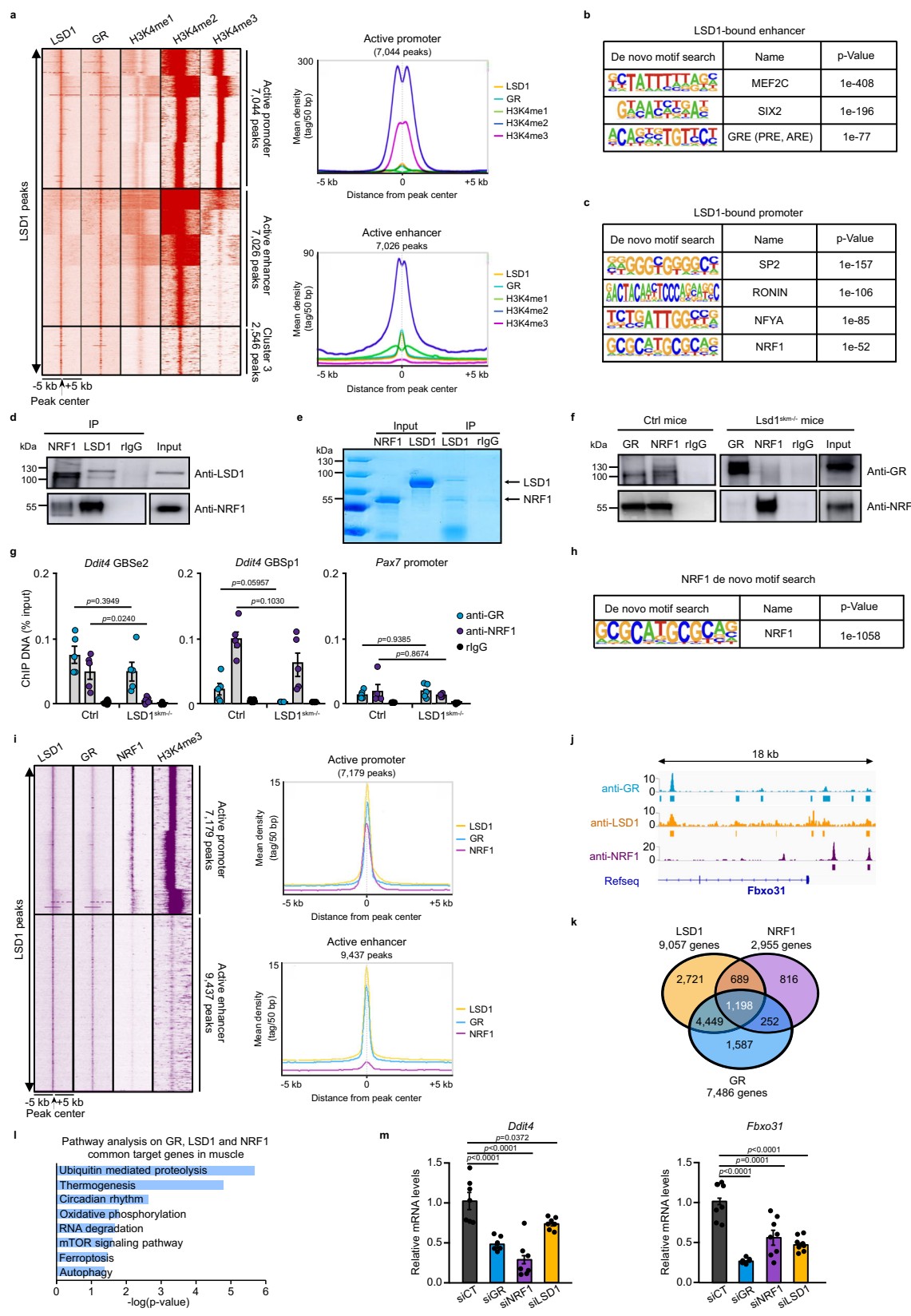

starvation of control mice, with loss of myofilaments, rupture of Z-lines and enlarged sarcoplasm, whereas less than 20% of the sarcomeres were damaged in muscles of LSD1(i)skm-/- and GR(i)skm-/- mice (Fig. 3i and Supplementary Fig. 3m). Taken together, these results show that myofiber GR and LSD1 are required for starvation-induced muscle atrophy.

To investigate the molecular mechanism by which LSD1 and GR promote starvation-induced muscle wasting, we determined their

**Fig. 2 | LSD1 bridges GR at active enhancers and NRF1 at active promoters to control target gene expression. a** Tag density map of LSD1, GR, H3K4me1, H3K4me2 and H3K4me3 in skeletal muscles, +/− 5 kb from the LSD1 peak center, and corresponding average tag density profiles. **b, c** HOMER de novo motif analysis of LSD1 binding sites located at enhancer (**b**) or promoter (**c**) regions. *p* value: hypergeometric testing. **d** Representative western blot analysis of NRF1 and LSD1 co-immunoprecipitation in gastrocnemius muscle nuclear extracts. Rabbit IgG served as a control for immunoprecipitation. *N* = 3 mice. **e** Representative SDS-PAGE of recombinant LSD1 and NRF1 proteins immunoprecipitated with anti-LSD1 antibodies or rabbit IgG as a control. Input corresponds to 10% of the purified LSD1 and NRF1 proteins. **f** Representative western blot analysis of GR and NRF1 co-immunoprecipitation in gastrocnemius muscle nuclear extracts from control and LSD1$^{skm-/-}$ mice. Rabbit IgG served as a control for immunoprecipitation. *N* = 3 mice. **g** ChIP-qPCR analysis performed with anti-GR and anti-NRF1 antibodies, or rabbit IgG in gastrocnemius muscles of ctrl and LSD1$^{skm-/-}$ mice at the GBSe2 and GBSp1 of *Ddit4*. The promoter region of *Pax7* was used as a negative control. *N* = 5 biological replicates. Mean ± SEM. One-way ANOVA with Tukey correction. **h** HOMER de novo motif analysis of NRF1 binding sites. *p* value: hypergeometric testing. **i** Tag density map of LSD1, GR, NRF1 and H3K4me3 in skeletal muscles, +/− 5 kb from the LSD1 peak center and corresponding average tag density profiles. **j** Localization of GR and LSD1 at the *Fbxo31* locus. **k** Overlap among genes bound by LSD1, GR or NRF1 in skeletal muscles. **l** Pathway analysis on LSD1, GR and NRF1 common target genes in skeletal muscles, with *p* values adjusted using the Benjamini-Hochberg. **m** Relative *Ddit4* and *Fbxo31* transcript levels determined in C2C12 myotubes transfected with siRNA directed against *Gr*, *Nrf1* and *Lsd1* (siGR, siNRF1 or siLSD1, respectively), or with a scramble siRNA (siCtrl). *N* = 7 biologically independent samples. Mean ± SEM. One-way ANOVA with Tukey correction. Source data are provided as a Source Data file.

cistrome 24 h following food deprivation. We observed an increased GR recruitment on chromatin after fasting (from 14,108 to 37,783 peaks, Fig. 4a, b and Supplementary Fig. 4a, b), and most genes bound by GR in fed condition were also bound upon fasting (Supplementary Fig. 4c). The genes to which GR was de novo recruited after food deprivation were associated with calcium signaling and proteasome system among other pathways (Supplementary Fig. 4d). Surprisingly, LSD1 recruitment to chromatin was markedly decreased after 24 hours of food deprivation, from 16,616 to 692 peaks (Fig. 4a, b, Supplementary Fig. 4a, b), despite similar LSD1 protein levels in skeletal muscles of mice fed and fasted for 12 to 48 h (Fig. 4c, d).

To determine the kinetics of GR and LSD1 recruitment during starvation, we performed ChIP-qPCR analysis at 0, 3, 6, 9 and 24-h of food deprivation. While the association of GR with GBSs at enhancers of anti-anabolic (*Pik3r1*) and catabolic (*Ddit4* and *Trim63*) genes was unaffected by food deprivation, LSD1 recruitment to these sites was increased at 3 h and subsequently decreased at 6 h (Fig. 4e), indicating that LSD1 plays a role at the early stage of starvation. Importantly, ChIP-qPCR analysis revealed that the abundance of the repressive histone mark H3K9me2 already reduced at 3 h of food deprivation, when LSD1 is still present at the chromatin (Fig. 4e). H3K9me2 levels remained low at the various loci beyond 6 h of starvation, indicating that LSD1 demethylates H3K9 within the first 3 h following food deprivation, leading to the derepression of its target genes (Fig. 4e). Of note, GR and LSD1 were not enriched at the *Pax7* negative control locus, and the levels of H3K9me2 were not modulated by food deprivation at this site. In line with these data, the abundance of H3K9me1 and H3K9me2 was reduced at LSD1-bound loci, following a 24-h food deprivation, as exemplified by the *Tfcp2*, *Mtch1*, *Calm* and *Rap2b* loci (Fig. 4a, b). Thus, these data strongly indicate that LSD1 demethylates H3K9 at genes associated with muscle atrophy within first 3 h of food deprivation to promote their transcription. In agreement, RT-qPCR analysis confirmed that the transcript levels of "atrogenes" were stimulated by more than 2-fold 12 h following food deprivation to reach 10-fold at 24 h and 20-fold at 48 h (Supplementary Fig. 4e).

Given that the LSD1 enzymatic activity depends on the metabolic co-factor FAD, we assessed FAD levels in muscle tissue at various times of food deprivation, and found that they declined by 30% at 3 h and by 50% at 24 h (Supplementary Fig. 4f), potentially leading to decreased LSD1 demethylase activity.

Altogether, our results provide evidence that the GR/LSD1 complex triggers the expression of starvation-induced proteolysis genes in skeletal muscles, at least in part via the demethylation of H3K9 at LSD1-bound genomic locations.

### The GR/LSD1 complex mediates DEX-induced muscle atrophy
To determine whether LSD1 also contributes to GR activity in the presence of pharmacological levels of synthetic GR ligands, mice were intraperitoneally injected with dexamethasone (DEX) at 10 mg/kg/day for 3 days. Such a treatment decreased fast-twitch muscle weight and

strength in control mice, in agreement with previous reports[27,30,31], but not in LSD1$^{(i)skm-/-}$ and GR$^{(i)skm-/-}$ littermates (Fig. 5a, b and Supplementary Fig. 5a, b). In agreement, the transcript levels of genes involved in autophagy and the proteasome system were induced in DEX-treated control muscles, but not in those where GR or LSD1 were depleted (Fig. 5c and Supplementary Fig. 5c). In addition, whereas DEX did not affect LC3I/II and p62 levels in muscles of control mice, we noticed an upregulation in protein degradation pathway activity, as revealed by decreased phosphorylation of FOXO3A and GSK3B (Fig. 5d, e and Supplementary Fig. 5d−f), associated with a reduced activity of anabolic regulators MTOR and 4E-BP1 (Fig. 5d, e and Supplementary Fig. 5d−f). This effect was not observed in DEX-treated LSD1$^{(i)skm-/-}$ and GR$^{(i)skm-/-}$ mice, showing that DEX-induced expression of genes involved in catabolism and DEX-impaired expression of genes involved in protein synthesis are myofiber GR- and LSD1-dependent.

Of note, GR and LSD1 protein levels decreased within 24 h of DEX treatment, but returned to basal levels at 72 h (Fig. 5f). At this time point, LSD1 interacted with GR and NRF1 in skeletal muscle nuclear extracts of mice treated with vehicle or DEX (Fig. 5g). GR and LSD1 binding to the enhancer GRBS of *Trim63* and *Ddit4* in skeletal muscles was increased after a 72 h DEX treatment of control mice, but not of LSD1$^{(i)skm-/-}$ mice (Fig. 5h), showing that LSD1 contributes to DEX-stimulated GR recruitment to cognate binding sites.

Together, our data show that LSD1 is required for GR transcriptional activity at pharmacological GC levels.

### LSD1 demethylase activity is required for GR-dependent muscle wasting, but is dispensable for glucocorticoid anti-inflammatory activity
To further investigate the molecular features of the GR/LSD1 complex, we analyzed LHCN-M2 human myotubes. Immunocytofluorescence analysis showed that LHCN-M2 cells express both GR and LSD1, and that the two factors are mainly located in the nucleus (Fig. 6a). Similar to what we observed in mouse muscles, co-immunoprecipitation experiment showed that GR, LSD1 and NRF1 interact in LHCN-M2 nuclear extracts (Fig. 6b).

In agreement with in vivo data, DEX treatment enhanced the transcript levels of genes involved in anti-anabolic and catabolic pathways in LHCN-M2 cells (Fig. 6c). To determine whether LSD1 activity is required for this induction, myotubes were treated with the LSD1-specific nanomolar affinity inhibitor CC-90011 (also named Pulrodemstat besylate, QC6688[36]). Our RT-qPCR data revealed that a co-treatment with CC-90011 abrogated DEX-dependent atrogene induction, thus showing that inhibition of LSD1 activity counteracts DEX effects in tissue culture. We thus aimed at characterizing the effects of this inhibitor in a mouse model.

To investigate whether pharmacological LSD1 inhibition impacts DEX-induced muscle atrophy, we analyzed mice co-treated for 72 h with DEX and/or CC-90011. Remarkably, whereas muscle strength and fast-twitched muscle mass were decreased by more than 20% in

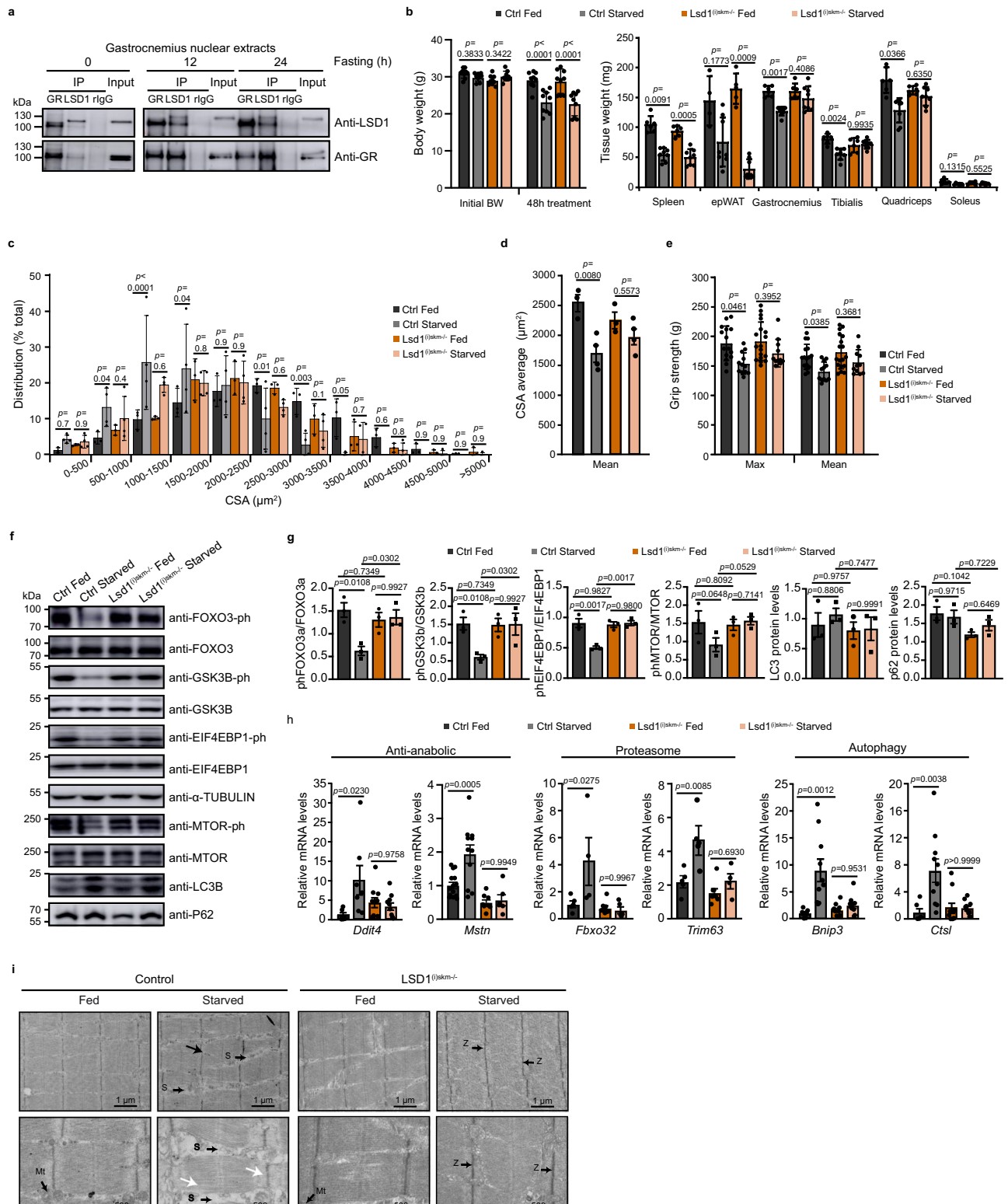

DEX-treated mice, they were decreased by less than 10% in mice co-treated with the inhibitor (Fig. 7a, b). GR recruitment to the enhancer of *Trim63* and *Ddit4* was reduced upon LSD1 inhibition (Fig. 7c), and RT-qPCR analysis revealed that DEX-induced transcript levels of genes involved in UPS and autophagy were blunted in the presence of the LSD1 inhibitor (Fig. 7d), showing that LSD1 demethylase activity is required to mediate DEX-dependent muscle wasting. Note that a 3-days treatment with CC-90011 alone had no major impact on muscle mass nor on the expression of GR target genes (Fig. 7a, b, d).

Importantly, spleen weight similarly decreased in response to DEX treatments, in the presence or absence of CC-90011 (Fig. 7a), and its immune cell composition was evenly impaired (Supplementary Fig. 6a, b). Moreover, DEX reduced the total blood count of leukocytes, which was mainly due to an 80% reduction in lymphocyte numbers, leading to a shift in the composition in white blood cells towards a higher percentage of neutrophil, that was not impacted by CC-90011 (Fig. 7e).

To further determine whether the treatment with CC-90011 affects DEX-induced anti-inflammatory activities, naive lymphocyte

**Fig. 3 | LSD1 is required for starvation-induced muscle atrophy. a** Representative western blot analysis of gastrocnemius muscle nuclear extracts from mice fed or starved for 12 or 24 h, immunoprecipitated with anti-GR or anti-LSD1 antibodies. Rabbit IgG served as a control for immunoprecipitation. $N = 3$ mice. **b** Body, gastrocnemius, tibialis, quadriceps, soleus, spleen and epWAT weights of 12-week-old Ctrl and LSD1$^{(i)skm-/-}$ mice fed or starved for 48 h. $N = 9$ mice. Mean ± SEM. Two-way ANOVA with Tukey correction. **c, d** Distribution (**c**) and average CSA (**d**) of gastrocnemius of 12-week-old Ctrl and LSD1$^{(i)skm-/-}$ mice fed or starved for 48 h. $N = 3$. Mean ± SEM. Two-way ANOVA with Tukey correction (**c**), one-way ANOVA with Tukey correction (**d**). **e** Maximal (Max) and average (Mean) grip strength of 12-week-old Ctrl and LSD1$^{(i)skm-/-}$ mice fed or starved for 48 h. $N = 16$ Ctrl fed, 12 Ctrl starved, 18 LSD1$^{(i)skm-/-}$ fed and 11 LSD1$^{(i)skm-/-}$ starved mice. Mean ± SEM. Two-way

ANOVA with Tukey correction. **f, g** Representative western blot analysis (**f**) and corresponding quantification (**g**) of the indicated proteins in quadriceps of 12-week-old Ctrl and LSD1$^{(i)skm-/-}$ mice fed or starved for 48 h. α-TUBULIN was used as a loading control. $N = 3$ mice. Mean ± SEM. One-way ANOVA with Tukey correction. **h** Relative transcript levels of indicated genes determined in gastrocnemius of 12-week-old Ctrl and LSD1$^{(i)skm-/-}$ mice fed or starved for 48 h. $N = 6$ Ctrl Fed, 10 Ctrl Starved, 10 LSD1$^{(i)skm-/-}$ Fed and 10 LSD1$^{(i)skm-/-}$ Starved mice. Mean ± SEM. One-way ANOVA with Tukey correction. **i** Ultrastructure analysis of gastrocnemius muscles of 12-week-old Ctrl and LSD1$^{(i)skm-/-}$ mice fed or starved for 48 h. Mt mitochondria, S sarcoplasm, Z Z line. Black arrow indicates Z line disruption; white arrows indicate loss of myofilaments. This experiment has been performed on a minimum of 5 mice per group. Source data are provided as a Source Data file.

T helper (Th) cells were isolated from lymph nodes of wild-type mice and polarized towards Th0 or Th17 conditions. Flow cytometry analysis confirmed that the proportion of IL-17 expressing cells was strongly enhanced in Th17 compared with Th0 cells (57% versus 1.2%, respectively). Notably, IL-17 production was 3-times reduced in cells treated with DEX, either alone or in combination with CC-90011 in Th17 cells (Fig. 7f), showing that LSD1 inhibition does not affect DEX anti-inflammatory activity in this cell type. Taken together, these results show that the LSD1 inhibitor CC-90011 limits DEX-induced muscle wasting without impairing anti-inflammatory activities in mice in physiological conditions.

### CC-90011 does not impair GC anti-inflammatory activities in a mouse model of colitis

To investigate the effects of LSD1 inhibition on GC anti-inflammatory effects in pathological conditions, we employed a mouse model of inflammatory bowel disease (IBD). Mice were treated with 3% dextran sulfate sodium (DSS) in drinking water over a six-day period, and DEX and/or CC-90011 were administrated from the 3rd to the 5th day of colitis, as described[37] (Supplementary Fig. 7a). As shown in Fig. 8a, b, water consumption and weight loss were similar for all DSS-treated mice. They all presented diarrhea and/or hematochezia, which was partially prevented by DEX and/or CC-90011 (Supplementary Fig. 7b, c). Importantly, the DSS-induced reduction of colon length was less prominent in mice treated with DEX, in combination or not with CC-90011, whereas CC-90011 alone had no effect (Fig. 8d, e). In addition, the induction of *IL-6* transcript levels in the colon of DSS-treated mice was reduced by treatment with DEX + /- CC-90011 (Fig. 8f). Importantly, whereas colon of mice treated with DSS presented an important infiltration of immune cells (red arrows) associated with a severe epithelial cell exfoliation at the apical pole of the crypts (black arrows), DEX + /- CC-90011 treatment markedly alleviated these damages (Fig. 8g and Supplementary Fig. 7c), thereby showing that CC-90011 does not impair DEX therapeutic effects on colon inflammation. Notably, mice treated only with CC-90011 presented an intermediate phenotype between DSS- and DSS + DEX-treated mice.

DEX treatment, both with and without CC-90011 also reduced DSS-induced spleen enlargement (Fig. 8h). It led to a shift in immune cell composition, characterized by decreased proportion of lymphocytes and in particular lymphoid CD4+ cells, an increased in CD11b+ cells mainly due to neutrophils, even though the number of monocytes and macrophages was reduced (Fig. 8i and Supplementary Fig. 7d). Notably, CC-90011 alone also prevented spleen hypertrophy, without affecting the proportion of immune cell populations (Fig. 8h, i and Supplementary Fig. 7d). Complete blood count (CBC) analysis revealed that all DSS-treated mice presented a severe normocytic anemia (Fig. 8j), most probably due to hematochezia. DEX + /- CC-90011 significantly reduced the number of lymphocytes by 70%, while increasing neutrophils and monocytes by 60%, whereas CC-90011 alone had no major effect on white blood cell proportions (Fig. 8j).

In agreement with Fig. 7a, even though the mass of fast-twitch muscles was decreased by DEX-treatment, co-treatment with CC-90011 reduced this diminution (Fig. 8k), thereby showing that CC-90011 does not impair DEX anti-inflammatory activity, while preventing GC-induced muscle atrophy.

Together, our data show that LSD1 inhibition impairs DEX-induced muscle wasting without affecting its anti-inflammatory activities in a model of acute inflammation.

## Discussion

GC-induced iatrogenic effects represent a severe clinical burden due to their propensity to induce muscle atrophy, thereby elevating the risks of falls and fractures. Despite extensive endeavors to synthesize GC analogs with anti-inflammatory properties dissociated from adverse effects, such compounds have not been obtained so far. Here we provide compelling evidence that LSD1 acts as a GR co-activator in mediating muscle atrophy, and that LSD1-specific inhibition counteracts muscle wasting provoked by GC without affecting their anti-inflammatory activities.

Our cistrome analyses uncovered about 16,000 LSD1 binding sites, with an even distribution across TSS, intronic and intergenic locations, as described in other tissues[36,38]. Taking advantage of genetically engineered mice, in which LSD1 is selectively ablated in myofibers, we demonstrate that LSD1 recruitment to these sites is highly myofiber-specific, and directly coordinates the expression of ~1600 genes. Importantly, we unveiled a cooperation between GR and LSD1, the latter being recruited to muscle enhancers in a spatially constrained domain centered around GR binding sites. The comparison between GR and LSD1 cistrome datasets showed that LSD1 is recruited to most genes targeted by GR. Moreover, transcriptomic analysis revealed that in the absence of either GR or LSD1, half of the genes bound by both factors is down-regulated. Collectively, these data demonstrate that LSD1 potentiates GR activity to control gene expression in myofibers. Recently, LSD1 has been shown to transactivate the androgen receptor (AR), another member of oxosteroid receptor family that coordinates muscle homeostasis[39]. AR/LSD1 interaction is enhanced when AR presents CAG (poly-Q) expansions causing spinobulbar muscular atrophy (SBMA)[39], suggesting that LSD1 exhibits distinct functional roles according to the oxosteroid receptor to which it is associated.

We previously showed that NRF1-bound promoters are topologically associated with GR-bound enhancers to regulate gene expression in mouse skeletal muscles[15]. In this study, we explored the interaction between GR, LSD1 and NRF1 in skeletal muscle gene regulation. ChIP-seq identified ~3000 NRF1 binding sites primarily at promoter regions linked to genes involved in proteolysis and oxidative phosphorylation. Furthermore, we provide evidence that LSD1 interacts with GR and NRF1, and that these interactions are required to stimulate gene expression, thereby establishing a functional link between GR at enhancer regions and NRF1 at promoter sites in myofibers. Using recombinant proteins, we provide evidence that the interaction

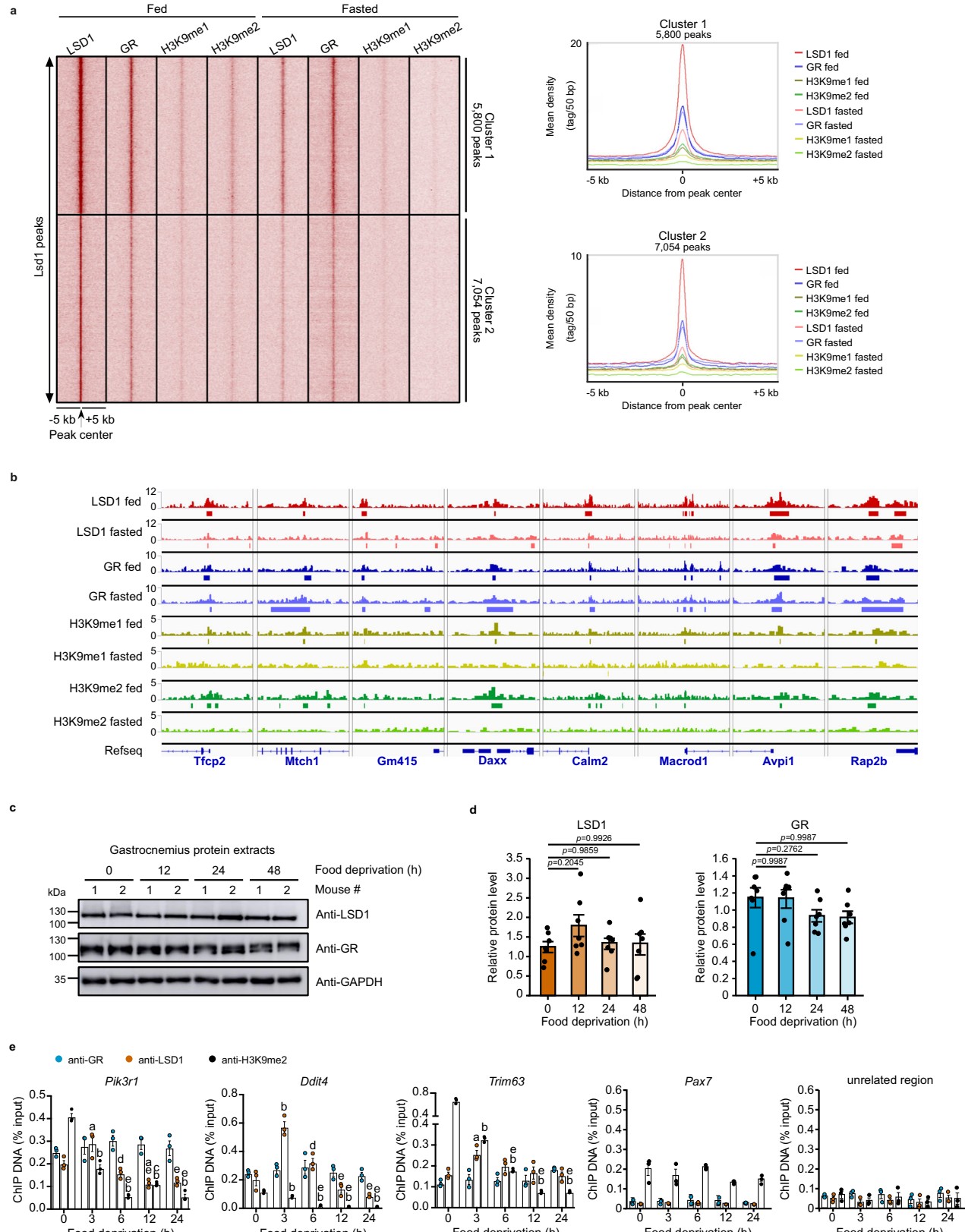

between LSD1 and NRF1 is direct. As it was previously shown that GR and LSD1 also directly interact, these data indicate that GR, LSD1 and NRF1 are part of a macromolecular complex that promotes gene transcription.

Our previous studies revealed that LSD1 cooperates with NRF1 to control metabolic properties of adipocytes, by promoting the expression of most genes encoding subunits of the mitochondrial respiratory chain, and of *Tfam*, the major regulator of mitochondrial biogenesis[38]. In muscles, OXPHOS and fatty acid metabolism pathways were not found as enriched gene networks upon GR and LSD1 ablation. Instead, we unveiled genes involved in muscle mass regulation, thereby showing that the LSD1/NRF1 complex holds specific functions

**Fig. 4 | The GR/LSD1 complex is required to trigger the expression of genes involved in starvation-induced proteolysis upon food deprivation. a** Tag density map of LSD1, GR, H3K9me1 and H3K9me2 in skeletal muscles of mice fed or fasted for 24 h, +/− 5 kb from the LSD1 peak center and corresponding average tag density profiles. **b** Localization of LSD1, GR, H3K9me1 and H3K9me2 in skeletal muscles of mice fed or fasted for 24 h at indicated loci. **c, d** Representative western blot (**c**) and corresponding quantification (**d**) of LSD1 and GR protein levels in gastrocnemius muscle extracts from 12-week-old wild type mice fed or food deprived for 12, 24 or 48 h. GAPDH was used as a loading control. $N = 2$ representative mice (**c**) out of 7 mice (**d**). Mean ± SEM. One-way ANOVA with Tukey correction. **e** ChIP-qPCR analysis performed at indicated loci with anti-GR, anti-LSD1 or anti-H3K9me2 antibodies in skeletal muscle of wild-type mice at 0, 3, 6, 12 and 24 h of food deprivation. The promoter region of *Pax7* and an unrelated region within the Ddit4 locus were used as negative controls. $N = 3$ mice. Mean ± SEM. One-way ANOVA with Tukey correction, with the following annotation on the figure. a: fed vs starved, $p < 0.05$. b: fed vs starved, $p < 0.001$. c: starved 3 h vs other time points, $p < 0.05$. d: starved 3 h vs other time points, $p < 0.01$. e: starved 3 h vs other time points, $p < 0.001$. Exact $p$ values are detailed in "Source Data". Source data are provided as a Source Data file.

according to the cell type and the transcription factors with which it is associated. Interestingly, our motif search indicates that LSD1 might interact with some other transcription factors, such as MEF2C or SIX2 at enhancer regions, suggesting that LSD1 might hold other functions in myofibers that remain to be determined.

Importantly, GR and LSD1 cooperate to control gene expression in skeletal muscles in the presence to both low and high GC levels, and the GR/LSD1 complex is required to promote fasting- and DEX-induced atrophy of fast-twitched muscle fibers. Even though GR and LSD1 are expressed and co-localize in every type of muscle from the limb, their glucocorticoid-dependent action on muscle wasting is clearly predominant in type-2 fibers, suggesting that an additional critical co-regulator is missing in slow-twitched muscles, which would be interesting to characterize.

Our results reveal that the molecular mechanism by which GR and LSD1 control gene expression differs upon starvation or pharmacological treatment. Both GR[(i)skm-/-] and LSD1[(i)skm-/-] mice are resistant to fasting induced muscle wasting. In addition, the major signaling pathways controlling protein degradation, such as ubiquitin proteasome system and autophagy, are similarly impaired in skeletal muscles of both GR[(i)skm-/-] and LSD1[(i)skm-/-] mice. Furthermore, as we provide evidence that GR and LSD1 interact during fasting, our data show that most pathways involved in fasting-induced muscle wasting and impaired by LSD1 ablation are GR-dependent. Muscle wasting can also be caused by additional pathways, including insulin or IGF1 signaling that activate AKT/MTOR cascade. Upon muscle atrophy, this signalization is impaired, thereby contributing to decreased protein synthesis. All these cascades are largely interconnected, as GR negatively regulates the AKT/MTOR pathway. However, since the starvation-induced loss of insulin signaling is not due to muscle defects, it is unlikely that myofiber LSD1 has any contribution in this process by interacting with additional transcription factors. ChIP experiments unveiled that food deprivation promotes GR binding to additional locations. Even though GR and LSD1 interact after a 24 h fast, LSD1 recruitment at GR-bound genes is impaired after 6 h of food deprivation. In the absence of LSD1 or GR, muscle mass was not reduced by food deprivation, showing the GR/LSD1 complex is crucial in promoting muscle wasting during food deprivation. Note that epididymal fat mass was even more reduced in LSD1[(i)skm-/-] and GR[(i)skm-/-] mice than in controls, most probably as a compensatory mechanism to maintain glucose levels. Together, these data show that LSD1 is required to promote GR-dependent muscle atrophy.

In response to DEX treatment, LSD1 and GR levels declined at 24 h, and went back to basal levels after 3 days. Similar observations were made for GR in spleen[31], suggesting a common DEX-dependent regulatory mechanism of GR expression in various tissues. Our data show that DEX promotes muscle wasting of fast-twitched myofibers in control mice, but not in GR[(i)skm-/-] mice, in agreement with previous reports[31], nor in LSD1[(i)skm-/-] mice, indicating that the GR/LSD1 complex is required for DEX-induced muscle atrophy. However, in contrast to food deprivation, GR and LSD1 binding to enhancer regions of target genes was induced after DEX treatment. Moreover, LSD1 ablation impaired DEX-induced GR recruitment at chromatin. Our genetic data

were further supported by pharmacological inhibition of LSD1. These results are in sharp contrast with those of Araki et al.[40], who recently reported that LSD1 loss enhances GC-induced muscle atrophy. They investigated LSD1 mutant mice in which exons 5 and 6 encoding for the C-terminal part of the SWIRM domain were deleted, potentially resulting in a C-terminal truncated protein encompassing the N-terminal, that could not be detected with the anti-LSD1 antibody ab17721, that is directed against LSD1 C-terminal region. Thus, as the N-terminal SWIRM region is key for the interaction with co-factors[41,42], a truncated LSD1 protein might have dominant negative effects. In contrast, in our study LSD1 ablation, determined using a N-terminal antibody (targeting aa 35-141), was assessed by deleting exon 1, leading to a frame shift in exon 2 that generates a stop codon, thereby preventing protein translation. An additional explanation for the observed discrepancies may originate from the overall design of the experiment. Indeed, the authors initiated the DEX treatment concomitants with the Tamoxifen administration for inducing *Lsd1* ablation. This approach presents two key issues, which are (1) an incomplete deletion of LSD1 protein and (2) a crosstalk between LSD1 and the two hormones[43–47]. Thus, the combination of our knock-out mouse models and the use of a LSD1-specific inhibitor reinforce our conclusions showing that LSD1 promotes GR-dependent gene expression.

In the presence of both natural and synthetic elevated GC levels, LSD1 demethylase activity is crucial for the action of the GR/LSD1 complex on gene expression. Even though LSD1 recruitment at GR-bound genes is reduced 6 h post-food deprivation, this time lapse is sufficient to demethylate H3K9 at anti-anabolic and catabolic genes, thereby promoting their expression. Interestingly, we found that the decrease in FAD content follows that of H3K9 methylation levels. Even though it is commonly admitted that FAD is required for LSD1 activity by binding to the amine oxidase-like (AOL) domain[17,48–50], to our knowledge, it was never investigated whether the presence of FAD is mandatory for LSD1 binding to the chromatin. We can thus speculate that upon starvation, FAD content becomes limiting, thereby affecting LSD1 folding and activity, ultimately leading to its release from the chromatin.

Previous studies have identified several LSD1 inhibitors, including CC-90011, Tranylcypromine, ORY1001 and GSK2879552[51,52]. CC-90011 binds rapidly and reversibly to the FAD cofactor within the LSD1 pocket and inhibits its activity, leading to changes in chromatin structure and gene expression[25]. IC50 of CC-90011 is of 0.25 nM, whereas Tranylcypromine irreversibly inhibits LSD1 with an IC50 value of 20.7 μM. This distinction underscores CC-90011 as a highly potent, selective, reversible and orally active LSD1 inhibitor. We show that CC-90011 strongly attenuates DEX-induced muscle wasting in mice, without impairing anti-inflammatory activities in physiological conditions and in a model of IBD. CC-90011 molecular function is achieved by preventing LSD1 recruitment to chromatin, thereby impacting GR transcriptional activity in muscle tissue. We also establish that DEX effectively alleviates colitis symptoms induced by DSS, confirming its anti-inflammatory efficacy. Notably, co-administration of the LSD1 inhibitor CC-90011 with DEX does not impair DEX anti-inflammatory effects, showing that LSD1 inhibition can selectively reduce GC-induced muscle atrophy without affecting their therapeutic benefits.

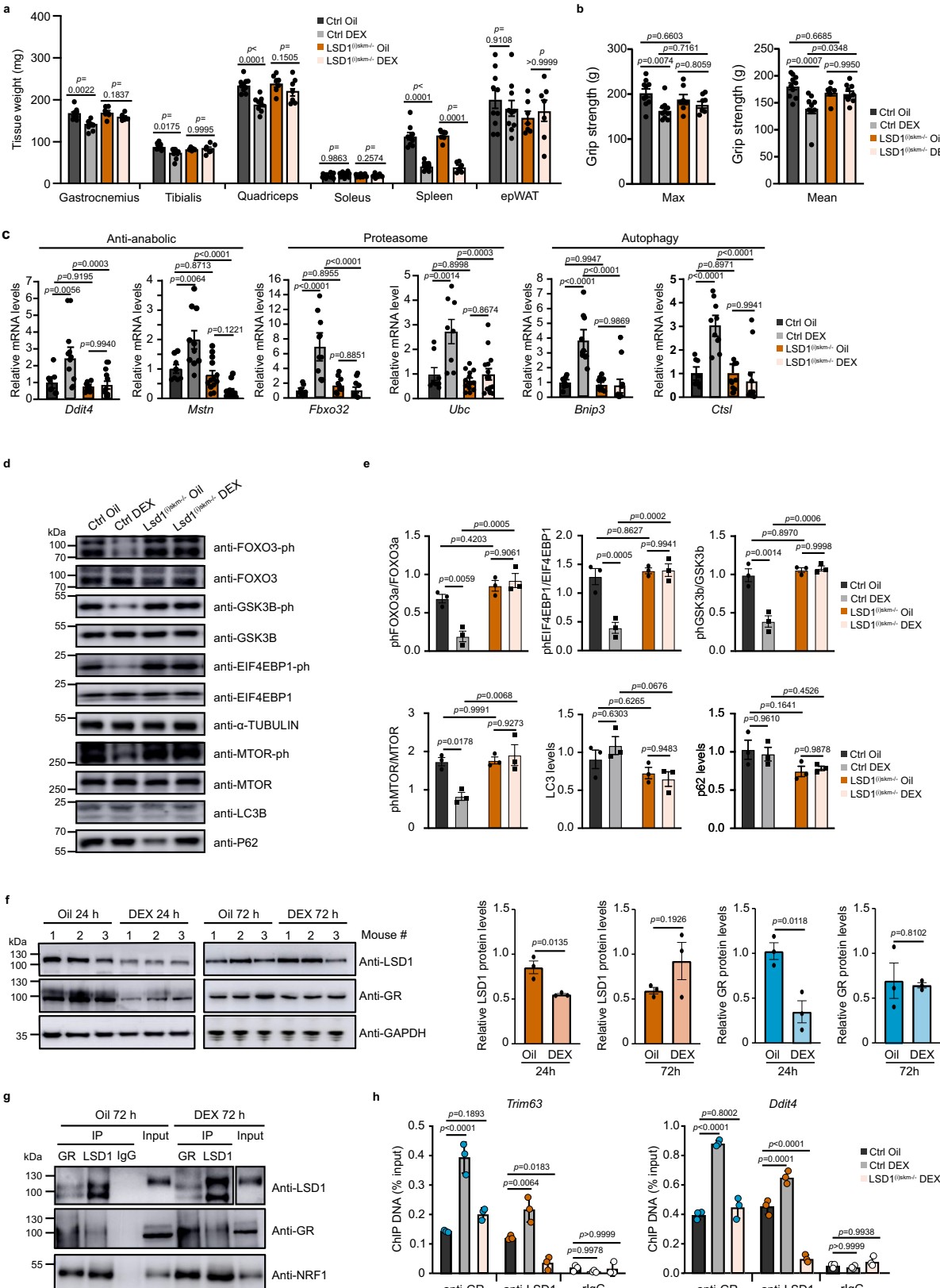

This finding is crucial for conditions like IBD, for which GC are the first line of treatment. Furthermore, we show that CC-90011 and DEX combination therapy does not adversely affect immune cell dynamics, indicating a promising approach to optimize inflammatory condition treatments by balancing anti-inflammatory benefits with muscle protection. It is to note that a treatment with CC-90011 alone limits DSS-induced spleen hypertrophy and slightly improves colon inflammation. Since CC-90011 is currently under clinical trials for AML, SCLC and solid tumor treatment[53], and counteracts DEX-induced catabolism of human myofibers, this compound is a promising candidate to attenuate GC muscular side effects in patients.

**Fig. 5 | LSD1 is required for dexamethasone-induced muscle wasting.**
**a, b** Gastrocnemius, tibialis, quadriceps, soleus, spleen and white epWAT mass (**a**), and maximal (Max) and average (Mean) grip strength (**b**) of 12-week-old Ctrl and LSD1$^{(i)skm-/-}$ mice treated with dexamethasone (DEX) or a vehicle (Oil) for 72 h. $N = 10$ Ctrl Oil, 10 Ctrl DEX, 7 LSD1$^{(i)skm-/-}$ Oil and 7 LSD1$^{(i)skm-/-}$ DEX mice. Mean ± SEM. Two-way ANOVA (**a**) and one-way ANOVA (**b**) with Tukey correction. **c** Relative transcript levels of indicated genes in gastrocnemius of Ctrl and LSD1$^{(i)skm-/-}$ mice treated with DEX or Oil for 72 h. $N = 4$ Ctrl Oil, 5 Ctrl DEX, 6 LSD1$^{(i)skm-/-}$ Oil and 7 LSD1$^{(i)skm-/-}$ DEX mice biological replicates, the individual values of the technical replicates are presented on the graph. Mean ± SEM. One-way ANOVA with Tukey correction. **d, e** Representative western blot (**d**) and relative levels of the indicated proteins (**e**) in quadriceps of control and LSD1$^{(i)skm-/-}$ mice treated with DEX or Oil for 72 h.

α-Tubulin was used as a loading control. $N = 3$ mice. Mean ± SEM. One-way ANOVA with Tukey correction. **f** Representative western blot analysis (left) and relative levels (right) of LSD1 and GR protein in gastrocnemius from three 12-week-old wild type mice treated with DEX or Oil for 24 or 72 h. GAPDH was used as a loading control. $N = 3$ mice. Mean ± SEM. Two-tailed $t$ test. **g** Representative western blot of gastrocnemius from mice treated with DEX or Oil for 72 h immunoprecipitated with anti-GR or anti-LSD1 antibodies. Rabbit IgG served as a control for immunoprecipitation. $N = 3$ mice. **h** ChIP-qPCR analysis performed at *Trim63* and *Ddit4* loci with anti-GR and anti-LSD1 antibodies or rabbit IgG in skeletal muscles of Ctrl and LSD1$^{(i)skm-/-}$ mice treated with DEX or Oil for 72 h. $N = 3$ mice. Mean ± SEM. One-way ANOVA with Tukey correction. Source data are provided as a Source Data file.

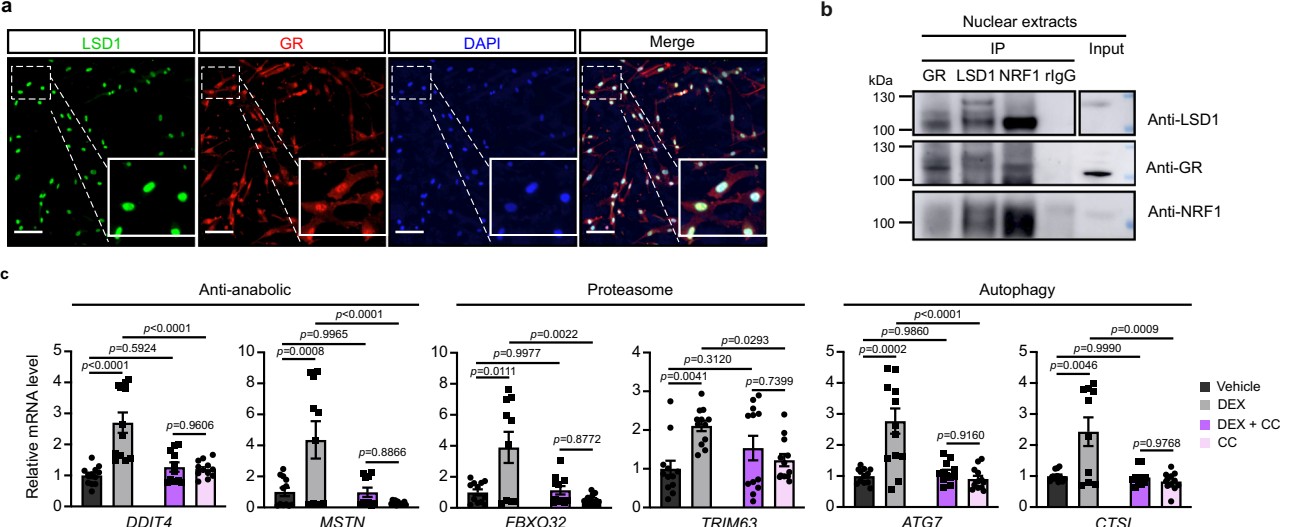

**Fig. 6 | Effect of LSD1 inhibition on dexamethasone-treated LHCN-M2 cells.**
**a** Representative immunofluorescent detection of LSD1 (green) and GR (red) in LHCN-M2 cells. Nuclei were stained with DAPI. Scale bar, 50 μm. $N = 3$ biological replicates. **b** Representative western blot analysis of nuclear or cytoplasmic extracts of differentiated LHCN-M2 cells, immunoprecipitated with anti-GR, anti-LSD1 or anti-NRF1 antibodies. Rabbit IgG served as a control for immunoprecipitation. **c** Relative transcript levels of the indicated genes determined in LHCN-M2 myotubes treated with vehicle, DEX, DEX with CC-90011 (DEX + CC) or CC-90011 (CC) for 24 h. $N = 6$ biological replicates, the 12 individual values of the technical replicates are presented on the graph. Mean ± SEM. One-way ANOVA with Tukey correction. Source data are provided as a Source Data file.

Together, by combining functional phenotypic and genome-wide analyses, we provide evidence that LSD1 acts as a GR co-activator to mediate GC-induced muscle atrophy. These findings shed insight into the molecular mechanisms underlying the iatrogenic effects of GC. Moreover, we demonstrate that pharmacological inhibition of LSD1 circumvents GC iatrogenic effects in skeletal muscles, while retaining their anti-inflammatory properties. Thus, our study opens perspectives to improve long-term GC treatments.

## Methods
### Mouse studies
To selectively ablate LSD1 in skeletal muscle fibers, LSD1$^{L2/L2}$ floxed mice, in which exon 1 is flanked with 2 LoxP sites[54], were intercrossed with HSA-Cre mice[55], to generate control (LSD1$^{L2/L2}$) and LSD1$^{skm-/-}$ mutant male mice. Alternatively, LSD1$^{L2/L2}$ mice were intercrossed with HSA-CreER$^{T2}$ mice[29], and seven-week-old LSD1$^{L2/L2}$ control male mice and HSA-CreER$^{T2}$/LSD1$^{L2/L2}$ sex-matched somatic pre-mutant littermates were intraperitoneally injected with Tamoxifen (1 mg/mouse/day) for 5 days to generate control (Ctrl) and LSD1$^{(i)skm-/-}$ mutant mice, respectively, as described[29]. GR$^{(i)skm-/-}$ mice were previously described[15]. All mice were on a C57Bl/6J background. Primers used for genotyping are listed in Supplementary Table 1.

Mice were maintained in a controlled temperature (19–23 °C) and humidity (40–60%) animal facility, with a 12-h light/dark cycle. Standard rodent chow (2800 kcal/kg, Usine d'Alimentation Rationelle, Villemoisson-sur-Orge, France) and water were provided *ad libitum*. Breeding and maintenance of mice were performed according to institutional guidelines. All experiments were done in an accredited animal house, in compliance with French and EU regulations on the use of laboratory animals for research. Intended manipulations were approved by the Ethical committee (Com'Eth, Strasbourg, France) and authorized by the French Research Ministry (MESR), conforming to the 2010/63/EU directive (APAFIS numbers: 2015-26, 37660, 39468 and 45167).

Food deprivation experiments were initiated one month after the last tamoxifen injection. Dexamethasone (Sigma; product D1756-1G) was dissolved at 20 mg/ml in EtOH, diluted at 3 mg/ml in oil, and intraperitoneally administrated at 10 mg/kg. LSD1 inhibitor CC-90011 was administrated *per os* at 5 mg/kg in 0.5% methylcellulose (Sigma; product M0512-110G; viscosity, 4000 centipoises [cP]). Both compounds were administrated one month after the last tamoxifen injection for one or three days.

Colitis was induced by 3% DSS (MP Biomedicals, France) added to the drinking water for 6 days. Daily assessments included water

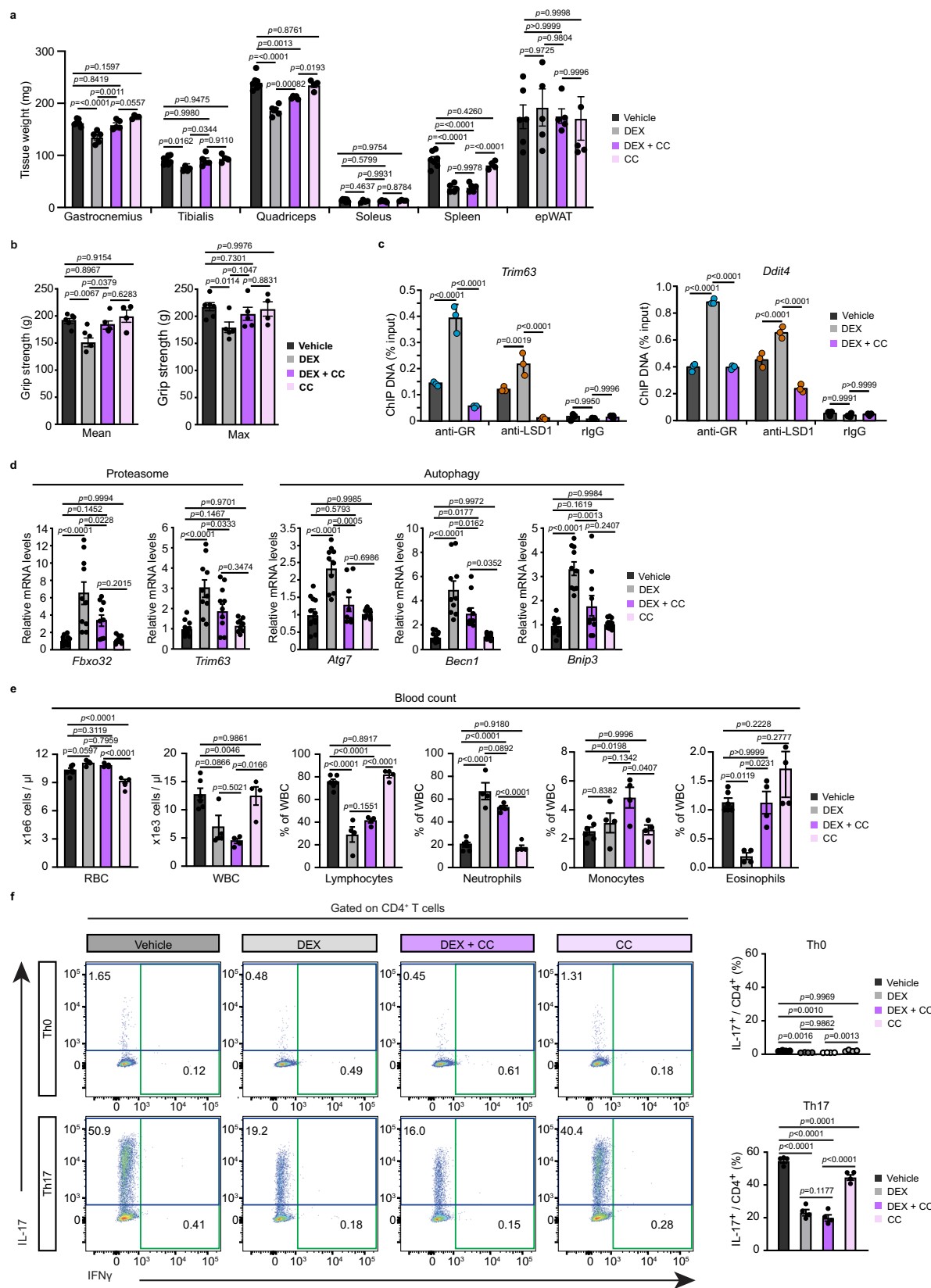

intake, body weight change, and clinical signs to calculate a clinical disease score. Each signal presented by the animal corresponded to one point and the sum of points for each mouse generated a clinical score. Dexamethasone and CC-90011 were administered daily from the 3rd to the 5th day of colitis at 10 mg/kg and 5 mg/kg, respectively.

CBC analysis was performed by the ICS phenotyping platform[56]. Limb grip strength was measured with a Grip Strength Meter (Bioseb),

**Fig. 7 | LSD1 inhibition prevents dexamethasone-induced muscle wasting without affecting glucocorticoid anti-inflammatory activities. a** Gastrocnemius, tibialis, quadriceps, soleus, spleen and epididymal white adipose tissue (epWAT) mass of 12-week-old wild-type mice treated with a vehicle, DEX, DEX with CC-90011 (DEX + CC), or CC-90011 (CC) for 72 h. *N* = 6 vehicle, 5 DEX, 5 DEX + CC, 4 CC mice. Mean ± SEM. Two-way ANOVA with Tukey correction. **b** Average (Mean) and maximal (MAX) grip strength of 72 h vehicle, DEX, DEX + CC or CC treated 12-week-old wild-type mice. *N* = 6 vehicle, 5 DEX, 5 DEX + CC, 4 CC mice. Mean ± SEM. One-way ANOVA with Tukey correction. **c** ChIP-qPCR analysis performed at *Trim63* and *Ddit4* loci with anti-LSD1 and anti-GR antibodies or rabbit IgG in skeletal muscles of wild-type mice treated with vehicle, DEX or DEX + CC for 72 h. *N* = 3 mice. Mean ± SEM. One-way ANOVA with Tukey correction. **d** Relative transcript levels of indicated genes determined in gastrocnemius muscles of 72 h vehicle, DEX, DEX + CC or CC treated wild-type mice. *N* = 5 biological replicates, the 10 individual values of the technical replicates are presented on the graph. Mean ± SEM. One-way ANOVA with Tukey correction. **e** Blood count analysis of wild-type mice treated with vehicle, DEX, DEX + CC or CC for 72 h. RBC: red blood cells, WBC: white blood cells. *N* = 6 vehicle, 4 DEX, 4 DEX + CC, 4 CC mice. Mean ± SEM. One-way ANOVA with Tukey correction. **f** Representative contour plots (right panel) and corresponding quantification (left panel) of IL-17 expression in Th0- or Th17-induced wild-type CD4 + T-cells treated with vehicle, DEX, DEX with CC-90011 (DEX + CC) and CC-90011 (CC). IFNγ was used as a control of Th17 induction. *N* = 4 mice. Mean ± SEM. One-way ANOVA with Tukey correction. Source data are provided as a Source Data file.

with three consecutive tests per session and recording mean and maximal values for each mouse[15].

Mice were euthanized by cervical dislocation, and tissues were immediately harvested, weighed, and either frozen in liquid nitrogen or processed for biochemical and histological analysis.

### Histological analysis

Hematoxylin and eosin staining and ultrastructural analyses were performed as described[57]. Images were acquired using a NanoZoomer S210 scanner (Hamamatsu) and a Mega View III camera (Soft Imaging System), respectively. Immunofluorescence analysis was performed as described[57,58] with anti-LSD1 (C-terminal, R. Schüle, #3544, 1:500) and anti-GR (Santa Cruz, sc393232, 1:500) antibodies. Rabbit and mouse IgGs (Santa Cruz, sc2357, sc 2025, 1:1000) were used as control. Sections were observed under Leica epifluorescence and confocal microscopes.

### Fiber cross-sectional area measurements

Muscle cross-sections were stained with dystrophin (Abcam, ab15277, 1:500) to mark the sarcolemma surrounding each fiber. CSAs were quantified using the FIJI image-processing software as described[58]. In brief, individual fibers were identified based on the intensity and continuity of the dystrophin-stained sarcolemma surrounding each fiber by segmentation. Areas were measured after background subtraction, automated thresholding and analyzed with the Qupath software.

### Flavin adenine dinucleotide (FAD) measurement

The assessment of FAD levels in skeletal muscles was conducted utilizing the FAD Assay Kit (ab204710, Abcam) in accordance with the manufacturer's guidelines. Briefly, skeletal muscles were lysed in ice-cold FAD Assay Buffer, and deproteinization with Perchloric acid/potassium hydroxide. FAD levels were determined through a colorimetric assay. FAD measurements were carried out concurrently and standardized to RNA concentrations, as ascertained by the NanoDrop spectrophotometer (Thermo Fisher).

### FACS analysis

Spleens were processed into a sterile 35 mm culture dish with PBS containing 1 mM EDTA, using a syringe plunger to crush the tissue. The cell suspension was filtered through a 70 μm strainer to remove clumps and washed with PBS. Cells were incubated for 10 min on ice with anti-CD45, anti-CD11b, anti−Ly-6G (Gr-1), anti-Ly-6C, and anti-F4/80 antibodies for the myeloid panel, or with anti-CD3ε, anti-CD4, anti-CD8a for the T-cells panel (Supplementary Table 4). Antibodies dilutions were prepared in DMEM (4.5 g/l glucose, 1% Penicillin/Streptomycin solution, and 2% bovine serum albumin, without phenol red). After washing, cells were analyzed using a BD LSR II flow cytometer and the FlowJo software.

### Cell culture

LHCN-M2 myoblasts[59] were kindly gifted by Dr. Jocelyn LAPORTE (IGBMC, Strasbourg University, France). Cells were grown in a 4:1 ratio of Dulbecco's modified Eagle medium (DMEM, Gibco, Cat # 10566016)/M199 (Gibco, Cat# 31150022) medium, supplemented with 15% heat-inactivated fetal calf serum (FCS), fetuin (25 μg/mL), human insulin (5 μg/mL), human EGF (5 ng/mL), human bFGF (0.5 ng/mL) and gentamycin (40 μg/mL). After reaching 80% confluency, cells were differentiated in DMEM (4.5 g/l glucose, 2% of heat-inactivated horse serum (HS), 1% Penicillin/Streptomycin solution) for 5 days. Cells were treated with CC-90011 (100 nM in DMSO) and/or DEX (100 nM in EtOH) for 24 h. Immunocytofluorescence assay was performed by incubating fixed LHCN-M2 cells with anti-LSD1 (R. Schüle, #3544, 1:500) and anti-GR (Invitrogen, MA1-510, 1:500) antibodies. Observations were made under a Leica confocal microscope.

C2C12 myoblasts (ATCC CRL-1772) were grown in DMEM (1 g/L glucose and 20% FCS). To induce myogenesis, the medium was switched to DMEM (1 g/L glucose and 2% HS) for 5 days. Cells were transfected with 30 pmol siRNA against GR (5′-GCUUUGCUCCU-GAUCUGAUUAUUAA-3′), Lsd1 (5′-CCCAAAGAUCCAGCUGACGUUU-GAA-3′), Nrf1 (5′-CCACACACAGUAUAGCUCAUCUCGU-3′), or a scrambled control (5′-AGGUUCCGUGUACGUAAGACAAACU-3′) (Invitrogen) using Lipofectamine RNAimax (Invitrogen) according to the manufacturer's instructions, two days before and one day after myogenic induction.

### RNA extraction and analysis

Muscles, C2C12 and LHCN-M2 cells were homogenized in TRIzol reagent (Life Technologies, Darmstadt, Germany). RNA was isolated using a standard phenol/chloroform extraction protocol, and quantified by spectrophotometry (Nanodrop, Thermo Fisher). 2 μg of total RNA underwent reverse transcription using SuperScript IV (Life Technologies) with oligo(dT) primers, according to the supplier's protocol. cDNA was diluted hundred times and quantitative PCR (qPCR) was performed with a Lightcycler 480 II (Roche) using the SYBR® Green PCR kit (Roche) according to the supplier's protocol (2 μl cDNA, 4.8 μl H2O, 5 μl Syber Green 2x mix and 0.2 μl of 100 μM primer mix). Primers are described in Supplementary Table 2. *18 S* and *RPLPO* were used as internal controls for mouse and human samples, respectively. Data were analyzed using the standard curve[60] and ΔΔCt[61] methods. Primer efficiency was calculated as Eff = 100*10^ ((−1/The Slope Value)−1).

For RNA-seq, RNA integrity was confirmed by Bioanalyzer. cDNA library was prepared, and sequenced with the standard Illumina protocol (HiSeq 2000, single-end, 50 bp) following the manufacturer's instructions. Image analysis and base calling were performed using RTA 2.7.7 and bcl2fastq 2.17.1.14. Adapter dimer reads were removed using DimerRemover (-a AGATCGGAAGAGCACACGTCTGAACTCCAGTCAC). FastQC 0.11.2 (http://www.bioinformatics.babraham.ac.uk/projects/fastqc/) was used to evaluate the quality of sequencing. Reads were mapped to the mouse mm10 genome (NCBI Build 38) using htseq-count (Version 0.9.1)[62]. Only uniquely aligned reads were retained for further analyses. Gene expression was quantified with HOMER. For comparison among datasets, transcripts with more than 50 raw reads were considered. Differentially expressed genes (DEGs) were identified using the Bioconductor libraries DESeq2[63] with a *p* < 0.05 and a fold change

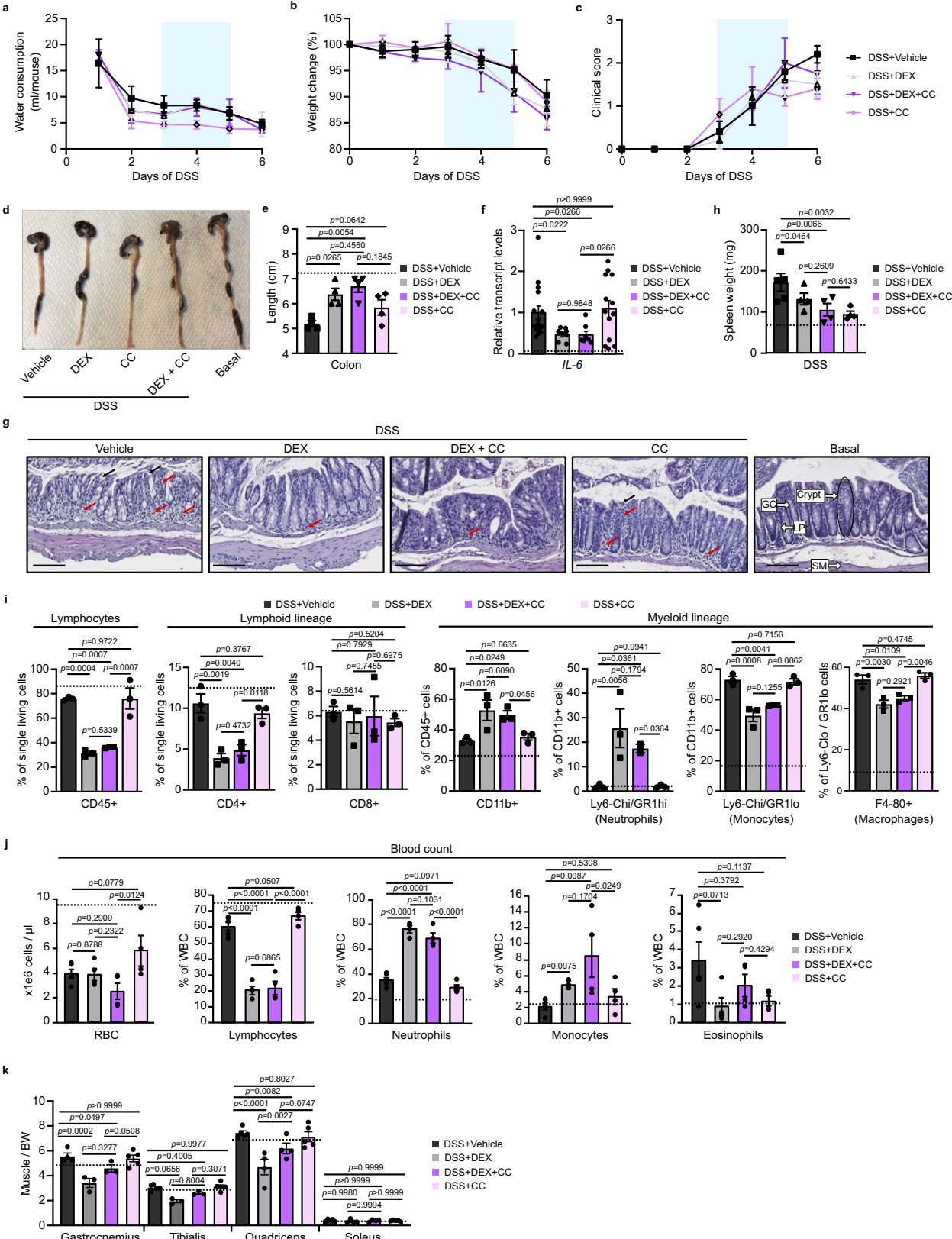

excluding values between 0.77 and 1.3. Pathway analysis was done using WebGestalt[64] using the Over-Representation Analysis (ORA) method and a $p < 0.05$. Heatmaps of normalized expression values were generated with Cluster 3.0[65] and MORPHEUS. Genes were clustered according to the hierarchical method (HCL clustering) using gene tree, the Pearson correlation and average linkage.

**Muscle nuclei isolation**

Nuclei were isolated from mouse skeletal muscles as described[66]. In brief, muscles were homogenized in hypotonic lysis buffer [10 mM HEPES-KOH pH 7.3, 10 mM KCl, 5 mM MgCl2, 0.1% NP-40, 0.1 M PMSF, protease inhibitor cocktail (45 μg/mL; Roche)]. For ChIP-qPCR and ChIP-seq experiments, lysates were fixed with 1% paraformaldehyde

**Fig. 8 | LSD1 inhibition does not impair dexamethasone-induced improvement of experimental colitis symptoms. a–c** Water consumption (**a**), weight change (**b**) and clinical score (**c**) of 12-week-old DSS-exposed wild-type mice treated with vehicle (DSS+vehicle), DEX (DSS + DEX), DEX with CC-90011 (DSS + DEX + CC) or CC-90011 (DSS + CC). $N = 5$ mice per condition. Mean ± SEM. Two-way ANOVA with Tukey correction. **d, e** Colon representative images (**d**) and length measurement (**e**) of mice exposed to indicated treatments. $N = 5$ mice per condition. Basal values are represented by a dashed line. Mean ± SEM. One-way ANOVA, correction for multiple comparisons. **f** Relative transcript levels of *IL-6* determined in colon. $N = 5$ mice per condition, all the individual values of the technical replicates are presented on the graph. Basal values are represented by a dashed line. Mean ± SEM. One-way ANOVA, correction for multiple comparisons. **g** Representative H&E staining of colons of mice from various groups. Red and black arrows denote inflammatory cells and exfoliated epithelial cells, respectively. GC goblet cell, LP lumina propria, SM muscularis mucosae composed of smooth muscle fibers. Scale bars, 100 μm. **h** Spleen weight of mice from various groups. $N = 5$ mice per condition. Basal values are represented by a dashed line. Mean ± SEM. Two-way ANOVA with Tukey correction. **i** Flow cytometry analysis of the lymphoid and the myeloid lineages from spleens of mice from various groups. $N = 3$ mice per condition. Basal values are represented by a dashed line. Mean ± SEM. Two-way ANOVA with Tukey correction. **j** Blood count analysis of mice from various groups. RBC: red blood cells. $N = 5$ mice per condition. Basal values are represented by a dashed line. Mean ± SEM. Two-way ANOVA with Tukey correction. **k** Gastrocnemius, tibialis, quadriceps and soleus mass relative to body weight (BW) of mice from various groups. $N = 5$ mice per condition. Basal values are represented by a dashed line. Mean ± SEM. Two-way ANOVA with Tukey correction. Source data are provided as a Source Data file.

(PFA) for 10 min, and neutralized with 125 mM glycine for additional 10 min. After further homogenization with a loose dunce and centrifugation at 1000 g for 5 min at 4 °C, pellets were resuspended in ice-cold hypotonic buffer, sequentially filtered through 70 μm and 40 μm strainers, and centrifuged at 1000 g for 5 min at 4 °C to collect the nuclei.

### Cell cytoplasm and nuclei protein extraction

Cells were lysed in 200 μL of Cytosolic buffer [10 mM HEPES, 60 mM KCl, 1 mM EDTA, 0.075% (v/v) NP40, 1 mM DTT, and 1 mM PMSF (pH 7.6)], and incubated on ice for 8 min. After centrifugation at 400 g for 5 min, supernatants were collected as cytosolic fractions. Nuclei pellets were resuspended in 100 μL of nuclear buffer [20 mM Tris HCl, 420 mM NaCl, 1.5 mM MgCl₂, 0.2 mM EDTA, 1 mM PMSF and 25% (v/v) glycerol (pH 8.0)], and incubated for 10 min on ice. After centrifugation at 15,000 g for 10 min, supernatants were collected as the nuclear fractions.

### Protein analysis

To isolate proteins, purified nuclei were resuspended in RIPA buffer [50 mM Tris, pH 7.5, 1% NP40, 0.5% sodium deoxycholate, 0.1% SDS, 150 mM NaCl, 5 mM EDTA and protease inhibitor cocktail (45 mg/ml, Roche, 11 873 580 001)] and incubated for 10 min at 4 °C.

For total muscle protein extraction, tissues were grounded in RIPA buffer at 4 °C, centrifuged at 12,000 g for 10 min, and supernatant was retained for further analyses.

For western blot analyses, homogenates were separated in polyacrylamide gels, transferred to Hybond nitrocellulose membranes (Amersham Biosciences), and probed with specific antibodies targeting LSD1 (C-terminal, R. Schüle, #3544, 1:1000), GR (C-terminal, IGBMC, #3249, 1:500), NRF1 (ab55744, Abcam, 1:1000), phospho-mTOR (Ser2448, Cell Signaling, 1:1000), mTOR (Cell Signaling, 1/500), phospho-4E-BP1 (Thr37/46, Cell Signaling, 1/1500), 4E-BP1 (53H11, Cell Signaling, 1:1500), phospho-FOXO3a (Ser318/321, Cell Signaling, 1:1000), FOXO3a (Cell Signaling, 1:1000), phospho-Akt (Thr308, Cell Signaling, 1:1000), Akt1 (2H10, Cell Signaling, 1:500), phospho-GSK3B (Ser9, 5B3, Cell Signaling, 1:1500), GSK3B (BD Transduction Laboratories, 1:1000), LC3B (GT1187, Genetex, 1:1000), P62 (ab56416, abcam, 1:5000), β-ACTIN (Santa Cruz, sc-4778, 1:5000), α-TUBULIN (IGBMC, 1Tub2A2, 1:5000), and GAPDH (#2118, Cell Signaling, 1:5000). Secondary antibodies conjugated to horseradish peroxidase (Jackson ImmunoResearch, 1:10,000) were detected using an enhanced chemiluminescence detection system (ECLplus, GE Healthcare) and an AI600 imager (GE Healthcare). Protein quantification was assessed by the FIJI/ImageJ distribution software (https://imagej.net/ImageJ)[67]. All uncropped blots and gels are presented in the Source Data file.

For Immunoprecipitation assays, 200 μg of muscle nuclear extracts were incubated with 5 μg of specific rabbit antibodies against GR (N-terminal, IGBMC, #3249), LSD1 (C-terminal, R. Schüle, #3544) or NRF1 (ab175932, Abcam), or control rabbit IgGs (Santa Cruz, sc2357) with Dynabeads protein G (Thermo Fisher Scientific, 10004D) in IP buffer, and processed for western bot analyses[68]. The loading volumes on the gel were adapted to optimize interaction visualization. In brief, the 40 μl obtained after IP of Ab1 were loaded as 10 μl for western blot detection with Ab1 and 30 μl for western blot detection with Ab2. Membranes were incubated with mouse anti-rabbit IgG (L27A9 Conformation Specific, Cell signaling, 1:5000) for 1 h at room temperature before addition of the secondary antibodies following manufacturer's instructions.

### Chromatin immunoprecipitation

ChIP followed by qPCR analysis (ChIP-qPCR) was performed on skeletal muscle nuclear extracts as described[66], using 5 μg anti-GR (C-terminal, IGBMC, #3249), anti-NRF1 (Abcam, ab175932), anti-LSD1 (C-terminal, R. Schüle, #20752) and anti-H3K9me2 (Active Motif, #39239), or a rabbit IgG (Santa Cruz, sc2357) negative control bound to protein Dynabeads protein G (Thermo Fisher Scientific, 10004D)[36]. Primers used for ChIP-qPCR are described in Supplementary Table 3.

For re-ChIP assays, after initial overnight immunoprecipitation, beads were washed, incubated in re-ChIP elution buffer for 30 min at 37 °C, and diluted 20-fold with ChIP dilution buffer supplemented with 50 μg BSA and protease inhibitor. A second immunoprecipitation reaction was performed with specific antibodies or control rabbit IgG. Protein–DNA complexes were eluted and reverse cross-linked for analysis[69].

For ChIP-seq analysis, libraries were prepared from 5 μg GR (C-terminal, IGBMC, #3249), LSD1 (C-terminal, R. Schüle, #20752), NRF1 (Abcam, ab175932), H3K9me1 (Active Motif, #39249) and H3K9me2 (Active Motif, #39239) immunoprecipitated DNA from skeletal muscle nuclear extracts as described[66]. ChIP-seq libraries were sequenced with an Illumina Hiseq 4000 as single-end 50 bp reads, and mapped to the mm10 reference genome using Bowtie 1.1.2[70]. Uniquely mapped reads were retained for further analysis. Reads overlapping with ENCODE hg38 blacklisted region V2 were removed using Bedtools[71]. Bigwig files were generated using Homer[72] software makeUCSCfile script with default parameters and scaled to 1e7 reads. MACS2 (2.2.7.1) algorithm (https://github.com/taoliu/MACS/)[73] was used for the peak calling and the appropriate input DNA from each sample was used as control. All peaks with an FDR greater than 0.01 were excluded from further analysis. The genome-wide intensity profiles were visualized using the IGV genome browser (http://software.broadinstitute.org/software/igv/)[74]. HOMER was used to annotate peaks and for motif searches[72]. De novo identified motifs were referred to as follow: R = purine (G or A); Y = pyrimidine (T or C). Genomic features (promoter/TSS, 5′ UTR, exon, intron, 3′ UTR, TTS and intergenic regions) were defined and calculated using Refseq and HOMER according to the distance to the nearest TSS. Clustering analyses were done with the seqMINER software[75], and clustering normalization was done with the K-Means linear option. Venn diagrams were generated with Venny (https://bioinfogp.cnb.csic.es/tools/venny/).

Pathway analysis was performed with WebGestalt using the Over-Representation Analysis (ORA) method[64].

Parameters were set as default, with the exception of the following: Bowtie (-m 1 -- strata -- best - y - S - l 40), MACS2 [callpeak -- gsize 1.87e9 -- nomodel -- extsize 150 -- broad --keep-dup auto], seqMINER (input bed files normalized to 20 million reads per sample).

ChIP-seq and ChIP-qPCR analyses have been performed on different muscle samples.

### In vitro interaction assays

LSD1 cDNA (encoding aa 2-852) was cloned into a pDEST8 vector with a C-terminal hexahistidine tag sequence, and protein was expressed in Sf21 insect cells using the baculovirus technology. NRF1 cDNA was cloned into pET28a vector (Twist Bioscience) with of a hexahistidine tag sequence and expressed in BL21 *Escherichia coli*. Cells were resuspended in lysis buffer (20 mM Tris-HCl, pH 7.5, 400 mM NaCl, 10% glycerol, 1 mM CHAPS, 0.1% Nonidet P-40, 1 mM TCEP, 10 mM imidazole), sonicated, and centrifuged at 10,000 g. Supernatants were processed through a HisTrap FF crude column (GE Healthcare), and proteins were eluted with 300 mM imidazole. Further purification was done using a Superdex S200 10/300 column (Cytiva) in a SEC buffer (50 mM Hepes, pH 7.5, 300 mM NaCl, 2 mM MgCl$_2$, 10% glycerol, 2 mM CHAPS, 0.1% Nonidet P-40, 1 mM TCEP). Protein purity and homogeneity were confirmed by SDS/PAGE.

For immunoprecipitation assays, Dynabeads protein G (Thermo Fisher Scientific, 10004D) were pre-cleaned with SEC buffer, and incubated with 5 μg anti-LSD1 antibody (R. Schüle, #3544) or control rabbit IgG (Pepro Tech, 500-P00-500 UG). Purified LSD1 (1 nmol) was added to the beads, followed by the addition 1 nmol NRF1. The mixture was incubated in SEC buffer at 4 °C for 3 h. Washing steps were carried out in SEC buffer with increasing amount of NaCl (400 mM to 1000 mM). Proteins were eluted in Laemmli buffer and analyzed on an 8% polyacrylamide SDS gel, with 10% of the purified proteins used as an input fraction.

### Th cells isolation and differentiation

Th17 polarization was performed as described[76] with minor changes. Antibodies information is detailed in Supplementary Table 5. Briefly, cells from peripheral and mesenteric lymph nodes of C57BL/6 mice were stained anti-CD16/CD32 blocking antibodies, anti-CD4, anti-CD8, anti-CD44, anti-CD25, anti-NK1.1, and anti-TCRgd antibodies in PBS with 10% heat-inactivated FCS for 15 min on ice. Naïve CD4 T cells (CD4 + , CD8-, CD44lo, TCRgd-, NK1.1-) were sorted using a FACS ARIA Fusion (BD Biosciences) with a purity >98%. Naive CD4 T cells (4×10⁴/well) were then activated with anti-CD3 (clone 2C11) and anti-CD28 (clone 37.51) antibodies, both pre-coated overnight in PBS at 4 °C on a nunc-immuno 96 well plate, with (Th17 conditions) or without (Th0 conditions) IL-6 (10 ng/ml) and TGFb1 (0.125 ng/ml) in the presence of neutralizing anti-IFNg and anti-IL-4 Abs (10 μg/ml each) in Iscove's Modified Dulbecco's Medium (IMDM) containing 10 % inactivated FCS, Glutamax, 10 mM Hepes, sodium pyruvate and Beta-mercaptoethanol (200 μl/well). After 3 days of culture in the presence of vehicle, DEX (100 nM) and/or CC-90011 (100 nM), cells were stimulated with phorbol-12-myristate-13-acetate (PMA) plus ionomycin (0.5 μg/ml each) and GolgiPlug (1/1000) for 2 h, stained with the BD HorizonTM fixable viability stain 780 Zombie, then with anti-CD4 antibody, fixed and permeabilized using the Intracellular Fix & permeabilization set (eBioscience) and stained with anti-IL-17 and anti-IFNg antibodies. Protein expression was analyzed on live CD4 + T cells using a flow cytometer symphony A1 (BD Biosciences).

### Statistical analysis

No statistical method was used to determine animal's sample size. Sample size was chosen based on experience with the used experimental models in the field of cell biology and animal experiments. The number of samples and independent biological experimental repeats are indicated in the figure legends. Data are represented as mean±SEM. Significance was determined using GraphPad Prism software (www.graphpad.com, GraphPad Software).

### Reporting summary

Further information on research design is available in the Nature Portfolio Reporting Summary linked to this article.

## Data availability

The raw and processed high-throughput sequencing datasets including RNA-seq and ChIP-seq data generated in this study have been deposited to the Gene Expression Omnibus (GEO) database under the accession number GSE230547. All remaining data is available in the Article, Supplementary and Source Data files. Source data are provided with this paper.

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

## Acknowledgements

We thank Dr. Laporte (IGBMC) for the kind gift of LHCN-M2 cells. We are grateful to Anastasia Bannwarth, Nikola Djordjevic, Jean-Marc Bornert, Laetitia Paulen, Régis Lutzing, and Joe Rizk for providing excellent technical assistance. We thank Nathalie Troffer-Charlier for the generation of the LSD1 baculovirus and for insect cell cultures. We thank the IGBMC animal house facility, the microscopy platform, the cell culture and molecular biology services, the flow cytometry facility, the histopathology service, the Mouse Clinical Institute (ICS, Illkirch, France), and the integrated structural biology and the GenomEast platforms. We are grateful to Michelina Plateroti, Bastien Launey and Matthieu Reslinger for sharing their expertise on IBD. We thank Dr. Susan Chan for sharing her expertise on flow cytometry analysis. This work was supported by funds from the Interdisciplinary Thematic Institute IMCBio, as part of the ITI 2021-2028 program of the University of Strasbourg, the Centre National pour la Recherche Scientifique (CNRS) and the Institut national de la santé et de la recherche médicale (Inserm), from IdEx Unistra (ANR-10-IDEX-0002), the French Infrastructure for Integrated Structural Biology (FRISBI) ANR-10-INSB-05 and Instruct-ERIC, and from SFRI-STRAT'US project (ANR 20-SFRI-0012) and EUR IMCBio (ANR-17-EURE-0023) under the framework of the French Investments for the Future Program. Additional funding was provided by INSERM, CNRS, Unistra, IGBMC, Agence Nationale de la Recherche (ANR-10-BLAN-1108, AndroGluco; ANR-16-CE11-0009, AR2GR; ANR-20-CE14-0040-01, LSD1GR; ANR-22-CE11-0014-01, MYOGLUCO), by an INSERM young researcher grant attributed to D.D., and by the grant ANR-10-LABX-0030-INRT, a French State fund managed by the ANR under the frame program Investissements d'Avenir ANR-10-IDEX-0002-02. Q.C. and S.S.C. were founded by the ANR-20-CE14-0040-01, V.U.-P. by the Ministère de l'Enseignement. Funding for open access charge: ANR.

## Author contributions

D.D., D.M., and Ro.S. formulated the initial hypothesis. Q.C., Ra.S., D.D., V.U.-P., F.T., F.C., S.S.-C., and G.L., carried out functional, molecular, and histological assessments of mice. Q.C., D.D., and T.Y. executed bioinformatics analyses. C.C. performed experiments on isolated Th cells. Ro.S. and E.M. provided LSD1 floxed mice, LSD1 antibodies, LSD1 expression vectors and LSD1 inhibitor. I.S. and I.M.L.B. performed biochemical analyses. Q.C., D.M., and D.D. took primary responsibility for data analysis and writing the manuscript.

## Competing interests

The authors declare no competing interests.
