## [Peer Review File · Nature Communications]

REVIEWER COMMENTS

Reviewer #1 (Remarks to the Author):

In this study, the authors outline a regulatory role for LSD1 as a co-activator of the glucocorticoid receptor (GR) in skeletal muscles. Using LSD1^{-/-} and GR^{-/-} mice models, they claim that LSD1 coordinates the interaction between GR and its enhancer protein, NRF1, to stimulate the downstream expression of genes under basal conditions. Further, they also state that LSD1 promotes GR-mediated atrogene expression upon starvation. Finally, they demonstrate that the GR-LSD1 complex may mediate dexamethasone-induced muscle atrophy in mice. There is considerable interest in therapeutic-based approaches to prevent muscle wasting. Thus, characterizing the role of a potential therapeutic target, namely LSD1, in DEX-mediated muscle atrophy would greatly interest the wider research community. Although the premise of the study is promising, there are various critical concerns, especially regarding the issues in the experimentation, data analysis, and lack of clarity on the mechanism.

Major comments:

1. I have concerns about the novelty of the manuscript. The role of LSD1 is demonstrated in the muscles (PMID: 20833138, PMID: 28228264, PMID: 36695573, and PMID: 29371665). A recent study also showed that the skeletal muscle-specific LSD1 loss exacerbated glucocorticoid-induced atrophy (Araki et al., eLife 2023), which is in contrast to the findings of this work. It is well-accepted that contrasting results are possible, but the author never attempted to explain the differences in the phenotype and the mechanisms found in this study.
2. The authors show that LSD1 is required for the interaction between GR and its enhancer protein, NRF1, to stimulate the downstream expression of genes under basal conditions. However, it is unclear how LSD1 may regulate NRF-1 to mediate GR activity in muscles. It would be worthwhile to explore the role of NRF1-dependent regulation of GR by LSD1 to gain a better mechanistic insight into the process.
3. In Fig 1C-E, the authors examine the GR and LSD1 genomic localization in myofibers and find that 80% of the DNA segments bound by LSD1 were also bound by GR. Are these DNA segments bound by NRF1 as well? This needs to be demonstrated as the manuscript also aimed to characterize the NRF1.
4. Previous studies have demonstrated that LSD1 regulates mitochondrial oxidative phosphorylation and fatty acid metabolism in muscles. I am wondering about the status of these genes in the model used in this study. Did the muscle-specific and whole-body LSD1^{-/-} mice exhibit changes in metabolism, especially in oxidative phosphorylation and fatty acid metabolism in muscles?
5. The authors have used starvation as a model to study whether LSD1 controls GR transcriptional activity. However, no data is shown whether the GR signaling is activated in this model. How were the authors convinced GR signaling is involved in this model, as starvation is known to induce multiple signaling pathways independent of GR in the muscle cells to induce atrophy?
6. The authors have used tamoxifen-dependent CreERT2 system-generated LSD1^{(j)skm}^{-/-} mice. Studies have shown that tamoxifen induces toxicity in muscles. How were the DEX injection experiments performed post-tamoxifen injection? The methods are unclear in the manuscript.
7. Muscle-specific and whole-body LSD1^{-/-} mice are the animal models used in this study. However, this mouse is not characterized well. It is essential to characterize the mouse model for various physiological parameters, including age-wise characterization of body weight, food, and water intake, blood profiling, etc. Also, characterize overall muscle phenotype in these mice under basal conditions by performing treadmill run experiments, metabolic cage experiments, and SDS COX staining at the least. In this work,

the authors may need to validate the muscle atrophy by examining the muscle fiber type upon dexamethasone treatment.

8. It is unclear whether LSD1 has a cell-autonomous role in skeletal muscle. The authors may perform in vitro experiments involving knock-down and overexpression of LSD1 using C2C12 or murine primary myotubes to rule out any possibility of paracrine and autocrine signaling influencing the muscle phenotypes in the mice models used for this study.

9. Numerous critical concerns exist with the immunoprecipitation experiments in the paper: The authors have used 10 % input, and most of the immunoprecipitated lanes have stronger bands than the input lane, which is inappropriate and impossible. Also, the authors have used single IgG control for two independent immunoprecipitation reactions performed with two different antibodies, which may also be wrong. Fig. 1B, 2D, 2F, 3A, 5F: It is unexpected to observe greater expression of the interacting partner protein than the immunoprecipitated protein. Please confirm if the origin of the secondary antibodies against LSD1 and GR primary antibodies is the same. Please verify the results and add loading controls to the panels. Note that, oddly, the input bands in the top and bottom panels are non-identical in Fig. 1B. In Fig. 4G: Are the IgG antibodies used for Trim63 and Ddit4 of different origins?

10. Critical controls are missing in the figures. In Figure 6, (A-E), extended figures 6 and 7B- the CC-90011 alone treatment group is missing. Without this group, it would be hard to interpret the data shown in Figures 6, extended Figures 6 and 7.

11. Fig. 1G-I: Given that GR is a major transcription factor for atrogens, it is crucial to reveal whether LSD1 interacts with the atroge loci in myofibres at physiological GC levels.

12. Fig. 6A, B: It is difficult to interpret the effect of LSD on DEXA-induced muscle atrophy without first ascertaining the impact of the LSD1 inhibitor itself on GR activity. It is strongly recommended to validate the inhibitor in the mouse model used.

13. Since the primary outcome of the paper is to characterize the use of LSD1 inhibitor to circumvent the muscle atrophy phenotype associated with the use of DEXA as an anti-inflammatory drug, it is crucial to observe the cytokine and chemokine profile in DEX-treated control and LSD^{-/-} mice, in addition to the immune cell population shown in the manuscript.

14. Fig. S1H-I: Please assess the extent and tissue specificity of LSD1 depletion in the mouse model by western blotting. The authors may test the expression of LSD1 in different muscles and organs.

15. Fig. 3: To assess the role of the GR/LSD1 complex in muscle atrophy, it is imperative to evaluate the activity of GR under starvation conditions in control and LSD^{skm}^{-/-} mice.

16. Fig. S4F: Given that FAD is depleted upon starvation, the enzymatic activity of LSD1 may be expected to reduce upon food deprivation. It is, therefore, important to investigate and comment upon the time-sensitive role of LSD1 in GR-mediated activation of starvation-induced muscle atrophy.

17. Fig. S1J, 3A, 3E, 4C: Throughout the paper, authors have characterized LSD function almost exclusively in the gastrocnemius muscle. Verifying these critical results in other important muscle types, especially the tibialis anterior (among other Type I and Type II muscle fibers), would be interesting.

18. Fig. 3E: Electron microscopy images are required to be quantified. Also, these results need to be verified by other methods.

19. Fig. 5D, E: Inconsistent use of loading controls in western blotting.

20. Fig. 4C, 5D-E: The authors have not quantified the western blots in this manuscript. All western blots need to be quantified and presented.

21. Extended figure 3C- fiber cross-section area (CSA) of GR(i)^{skm}^{-/-} mice starved for 48 h is much higher than the GR(i)^{skm}^{-/-} mice fed animals, which I feel is impossible. Fig. S3C: This figure indicates that

starvation induces hypertrophy in the muscles of GR^{-/-} mice. This is highly improbable since starvation is well known to inhibit protein synthesis. Furthermore, Fig. S3H indicates upregulation in mRNA expression of atrogenes, contradicting the hypertrophy phenotype reported.

Minor comments:

1. Please ensure uniform color coding across data figures in the manuscript.
2. Fig. 1H, S1E: Since the study is focused on skeletal muscles, please avoid displaying cardiac-associated genes in the list to avoid confusion.
3. Fig. 6, 7B: Please denote groups as Vehicle, DEX, and DEX+CC in the figure for simplicity.

Reviewer #2 (Remarks to the Author):

In this manuscript, Cai and colleagues investigate the molecular details underlying glucocorticoid-induced skeletal muscle atrophy, which limits the use of synthetic GC in patients that require chronic administrations. They report that LSD1 interacts with both GR and NRF1 to regulate gene expression. Pharmacological inhibitors of LSD1 attenuate muscle atrophy, providing evidence for the involvement of LSD1 and therapeutic promising applications. The number of reports that establish a role for LSD1 in skeletal muscle is increasing, so is the tight relationship with steroid hormone receptors. The manuscript is very well written, results are clear and conclusions are appropriate. Before publication, we have the following suggestions that may improve to strength the authors' conclusions.

- 1) In Fig. 1 the authors show colocalization of LSD1 and GR but not all the nuclei are positive, can the author stain for the different fiber types with MyHC marker subtype to establish whether there is a preferential expression of type IIb vs intermediate or oxidative fibers? Is there a homogeneous distribution of double positive nuclei in the entire transversal section, as the fiber type composition of gastrocnemius is not uniform.
- 2) In the IP experiment, the lower band of LSD1, that also the authors notice, is interesting, could that be another isoform (alternative splicing) or another member of the KDM family that interacts with GR? IP with an anti-GR Ab pulls down a lower band that is not visible in the input, may be because it is not enriched enough, but the upper isoform is not visible. This is problematic. However, CHIP analysis shows interaction at the chromatin level. But the authors should clarify why the bands have different size, or reconsider the interpretation of the IP experiments. LSD1 has isoforms (2a, 8a, etc).
- 3) Were the CHIP on target validations performed on different samples as those used for CHIPseq? If so, this strengthens the validation analysis.
- 4) The first sentence of the figure captions may be more informative with respect to "Characterization of...", "Role of..." can it be a sentence summarizing the main message of the figure?
- 5) Is the interaction of GR with LSD1 and NRF1 modulated by ligand? Can this be tested in cells treated with vehicle or GC?
- 6) The observation that LSD1 is required for interaction of GR with NRF1 (Fig 2F) suggests a specific effect, is it possible that LSD1 demethylates GR? Or is this effect (fig 2G) occurring at the chromatin level?
- 7) I suggest to move to main figure 3 analysis of CSA in control and starved mice with the relative quantification. Relative to autophagy, WB analysis of p62 and LC3 I/II should be added to conclude that autophagy is not induced and to determine is the autophagy flux is regular.

- 8) What would be the benefit to use CC-90011 with respect to TCP, for instance?
- 9) The effect of the inhibitor (occurring through the catalytic activity) implies that LSD1 demethylase activity is important, is it only involved in histone modification or is it the result of direct modification of GR or NRF1?
- 10) And what is the involvement of NRF1 in muscle atrophy? What do the data shown in Fig 2 add to the rest of the story?
- 11) The discussion and the manuscript would benefit if the authors take into consideration recent findings that establish a role of LSD1 in muscle in the context of another steroid receptor, androgen receptor. Especially considering the anabolic effects of androgen signaling in muscle. LSD1 represses transcription of several genes, but it transactivates AR, is this the case for GR? Similar specific effect that boosts GR transactivation?
- 12) For statistical analysis, ANOVA (not t test) shall be used for more than 2 sample comparisons.

Reviewer #3 (Remarks to the Author):

The study by Cai and collaborators investigates the relationship between the lysine-specific demethylase LSD1 and the glucocorticoid receptor (GR) in the context of muscle atrophy. The Authors show that these molecules directly interact resulting in the modulation of target gene expression. Among others, genes involved in the regulation of muscle mass homeostasis are targeted by LSD1 and GR interaction. The study is mainly based on genetic approaches, also taking advantage of mice in which the expression of GR or LSD1 is down-regulated. The results showing the interaction between LSD1 and GR are quite convincing, while few notes of care arise as for the relevance of such interaction to muscle atrophy, which is investigated in mice exposed to fasting or administered dexamethasone (DEX). Specifically, the following issues should be taken into consideration:

- line 85-86: while it is conceivable that some GRs are located in the nucleus also in physiological conditions such as those analyzed by the Authors, Figure 1A shows no or very little cytoplasmic GR, while this latter should be its main location in this experimental setting;
- one of the main findings reported in the present study is that muscle-specific lack of LSD1 confers a significant resistance to wasting induced in mice (age? initial body weight?) by 48 h fasting. Such a pattern is associated with inhibition of the expression of genes involved in catabolic pathways such as muscle-specific ubiquitin ligases and autophagy and with preservation of muscle ultrastructure. However, these results do not rule out that other pathways in addition to the GR-dependent one(s) could be altered by the lack of LSD1 in the skeletal muscle. In this regard, 48 h fasting results in a severe substrate shortage. While adipose tissue reduction, more marked in LSD1^{-/-} mice than in controls, likely attempts to compensate such a shortage, it is hardly able to achieve a total 'buffer' against the lack of nutrients;
- it would have been interesting to see the effects of LSD1 inhibitor in fasted animals;
- the second main observation of the study is that LSD1 down-regulation protects mice against DEX-induced atrophy, apparently without affecting DEX anti-inflammatory properties. However, only indirect evidence is provided in this regard, showing that spleen changes in size and both spleen resident and circulating immune cell sub-populations do not differ in the presence or in the absence of the LSD1 inhibitor. In this regard, the lack of effect on the anti-inflammatory properties should be demonstrated

using the LSD1 inhibitor on models such as the experimental autoimmune encephalomyelitis or inflammatory bowel disease;

- control mice treated with the LSD1 inhibitor are not included in the experimental design, while they should;
- LSD1 lack of action on the anti-inflammatory activity does not really fit with data reported in Figure 1I, which shows an up-regulation of genes involved in the Inflammatory Response Pathway;
- most of the results obtained in starved or DEX-exposed mice refer to gene expression data. However, since changes of mRNA levels do not necessarily match with protein expression and activity, at least some of the markers of protein hypercatabolism should be assessed at the protein level;
- data on human myotubes do not add any particular cue, they could be cited as supplemental;
- it not correct to talk about 'muscle catabolism' or 'muscle degradation'. Indeed, a tissue per se cannot be degraded or synthesized, rather its components (protein, lipids, etc.) are processed during anabolic/catabolic reactions. Similarly, it is not correct to talk about muscle atrophy when referring to an in vitro system (line 56);
- finally, in lines 203-204 LSD1 is reported to play a role in 'initiating the food deprivation process', which is a non-sense, since such a process starts when mice are no more allowed to reach food.

Reviewer #1 (Remarks to the Author):

In this study, the authors outline a regulatory role for LSD1 as a co-activator of the glucocorticoid receptor (GR) in skeletal muscles. Using LSD1^{-/-} and GR^{-/-} mice models, they claim that LSD1 coordinates the interaction between GR and its enhancer protein, NRF1, to stimulate the downstream expression of genes under basal conditions. Further, they also state that LSD1 promotes GR-mediated atrogene expression upon starvation. Finally, they demonstrate that the GR-LSD1 complex may mediate dexamethasone-induced muscle atrophy in mice. There is considerable interest in therapeutic-based approaches to prevent muscle wasting. Thus, characterizing the role of a potential therapeutic target, namely LSD1, in DEX-mediated muscle atrophy would greatly interest the wider research community. Although the premise of the study is promising, there are various critical concerns, especially regarding the issues in the experimentation, data analysis, and lack of clarity on the mechanism.

We thank the reviewer for her/his time and expertise in reviewing our submission, and are grateful for her/his constructive feedback. Note however that number of concerns were already discussed in the original version of the manuscript (see below).

Major comments:

1. I have concerns about the novelty of the manuscript. The role of LSD1 is demonstrated in the muscles (PMID: 20833138, PMID: 28228264, PMID: 36695573, and PMID: 29371665). A recent study also showed that the skeletal muscle-specific LSD1 loss exacerbated glucocorticoid-induced atrophy (Araki et al., eLife 2023), which is in contrast to the findings of this work. It is well-accepted that contrasting results are possible, but the author never attempted to explain the differences in the phenotype and the mechanisms found in this study.

In response to the concerns pertaining to the novelty of our findings, LSD1's role has been mainly studied in myogenesis (PMID: 20833138: Choi et al 2010 role of LSD1 in C2C12 myoblast differentiation, PMID: 28228264: Scionti et al. 2017 role of LSD1 in early muscle differentiation, and PMID: 29371665: Tomic et al. Lsd1 regulates skeletal muscle regeneration and directs the fate of satellite cells). While in the study of Araki et al. (2023, PMID: 36695573) LSD1 defines the fiber type-selective responsiveness to environmental stress by considering its interaction with FOXK1 in fast fibers and with ERRg in slow fibers, our manuscript offers unique findings in several ways: 1) beyond emphasizing LSD1's function as a GR co-activator, we dissect its intricate molecular mechanism in orchestrating glucocorticoid-induced muscular atrophy, 2) our genome-wide analyses provide comprehensive evidence of LSD1's functional importance, mapping out about 16,000 LSD1 binding sites that cooccur with those of GR at enhancers and those of NRF1 at promoters, 3) we have detailed how LSD1 is required for GR-mediated muscle atrophy in starvation and pharmacological conditions, which provides a broader insight into the multifaceted roles of LSD1 in skeletal muscles.

As mentioned in the original version of the discussion (page 14, lines 314-322, now page 18, lines 420-432), we provide possible explanations for the discrepancies with the study of Araki and colleagues (PMID: 36695573). Indeed, the mouse models used in both studies differ substantially. Our model was generated by deleting the exon 1 of *Lsd1*, leading to an early frame shift that prevents protein translation. In contrast, Araki et al. investigated LSD1 mutants in which exons 5 and 6 were deleted, which might result in a C-terminal truncated protein lacking the MAO domain and the TOWER domains. Since the authors used an antibody directed against LSD1 C-terminal domain, LSD1 truncated protein could not have been detected but might have had dominant negative effects, especially given the pivotal role of the N-terminal SWIRM domain in facilitating interactions with transcription factors and ancillary co-factors. Moreover, the time and length of TAM and DEX treatment also differ with the study of Araki and colleagues, in which DEX was administered concomitantly with TAM, thereby causing two main issues: 1) LSD1 might have been not fully depleted when DEX was applied, thus contributing to the promotion of atrogene transcription, and 2) altered estrogen receptor activity might interfere with that of GR when their ligand are applied simultaneously. In contrast, in our study, DEX was administered one-month after TAM injection. In addition, we show that both LSD1 ablation in skeletal muscle and LSD1 pharmacological inhibition counteracts DEX-induced muscle atrophy.

Altogether, we believe that our findings significantly enhance the knowledge on LSD1's function in skeletal muscles, and provide a strong foundation for future research to improve glucocorticoid treatments by minimizing muscle atrophy.

2. The authors show that LSD1 is required for the interaction between GR and its enhancer protein, NRF1, to stimulate the downstream expression of genes under basal conditions. However, it is unclear how LSD1 may regulate NRF-1 to mediate GR activity in muscles. It would be worthwhile to explore the role of NRF1-dependent regulation of GR by LSD1 to gain a better mechanistic insight into the process. In response to this concern, our RNA-seq analyses revealed that LSD1 loss in myofibers does not affect *Nrf1* and *GR* transcript levels. Moreover, as we stated in the discussion (page 14, lines 277-285, now page 16, lines 377-385), we demonstrate that LSD1 directly interacts with GR at enhancer regions and with NRF1 at promoters, and that these interactions are required to connect GR and NRF1, and stimulate gene expression by demethylating the H3K9me2 repressive mark. Indeed, our co-IP (Fig. 2f) and ChIP-qPCR experiments (Fig. 2g) demonstrate that LSD1 is essential for the optimal interaction between GR and NRF1 under basal conditions to promote target gene expression. These cooperative interactions establish a functional link between GR at enhancer regions and the transcription factor NRF1 at promoter sites in myofibers. In addition, our previous study revealed that GR recruitment to target genes was decreased by 50 % upon *Nrf1* knock-down, and that *Nrf1* silencing decreased the transcript levels of the GR target genes *Eif4ebp2* and *Pik3r1* by 50 % without impacting GR expression (doi: 10.1093/nar/gkab226), thereby showing that NRF1 is required for GR-dependent transcription. These datasets are further supported by our new dataset showing that GR, LSD1 and NRF1 silencing decreases the expression of GR target genes (new Fig. 2m and S2i).

3. In Fig 1C-E, the authors examine the GR and LSD1 genomic localization in myofibers and find that 80% of the DNA segments bound by LSD1 were also bound by GR. Are these DNA segments bound by NRF1 as well? This needs to be demonstrated as the manuscript also aimed to characterize the NRF1. To address this point, we have conducted a ChIP-seq analysis on NRF1 (new Fig. 2h-l and S2e-g). In agreement with our hypothesis, our studies reveal that the presence of GR and LSD1 at promoters strongly correlates with that of NRF1, in contrast with GR/LSD1 bound enhancers where NRF1 signal was not detectable (new Fig. 2i, j and S2g). These data were validated by independent ChIP-qPCR experiments at various loci (Fig. 2g and new S2h). These results further strengthen our understanding of the tripartite interactions at the genomic level (pages 7-8 lines 167-182).

4. Previous studies have demonstrated that LSD1 regulates mitochondrial oxidative phosphorylation and fatty acid metabolism in muscles. I am wondering about the status of these genes in the model used in this study. Did the muscle-specific and whole-body LSD1^{-/-} mice exhibit changes in metabolism, especially in oxidative phosphorylation and fatty acid metabolism in muscles?

To our knowledge, there is no study describing LSD1 role on mitochondrial oxidative phosphorylation and fatty acid metabolism in muscles of adult mice. This topic has mainly been addressed in either a developmental context, in fat tissue or in cancer cells (doi: 10.1182/bloodadvances.2020003521, 10.1038/ncomms5093, 10.1016/j.celrep.2016.09.053), which was addressed in the original version of the discussion (page 13, lines 286-292, now page 17, lines 388-394).

In this manuscript we analyzed either constitutive (LSD1^{skm^{-/-}}) or TAM-inducible (LSD1^{(i)skm^{-/-}}) muscle-specific LSD1 knock-out mice, as whole-body LSD1^{-/-} mice die during embryonic development (<https://doi.org/10.1038/ng.268>, doi: 10.1101/gad.2008511, doi: 10.1128/MCB.00521-10, doi: 10.1038/nature05671). The analysis of our RNA-seq data reveals that LSD1 ablation in myofibers does not affect the expression of genes involved in mitochondrial oxidative phosphorylation and has no major influence on fatty acid metabolism-related genes (Fig. R1), in contrast to what we previously described in the fat tissue where LSD1 control the vast majority of genes involved in OXPHOS and oxidative metabolism (DOI: 10.1038/ncomms5093). Instead, our data reveal that LSD1 is rather associated with immune response (Fig. R2). Additional pathway analysis of LSD1 direct targets is now presented in new Fig. S1n.

Figure R2: GSEA analysis of genes differentially expressed in gastrocnemius muscle of 9-week-old control and *LSD1^{skm}-/-* mice. Pathways related to upregulated genes are presented in orange, when those related to down-regulated genes are in blue.

5. The authors have used starvation as a model to study whether LSD1 controls GR transcriptional activity. However, no data is shown whether the GR signaling is activated in this model. How were the authors convinced GR signaling is involved in this model, as starvation is known to induce multiple signaling pathways independent of GR in the muscle cells to induce atrophy?

In response to food scarcity, mammals induce a hypothalamic-pituitary-adrenal (HPA) stress response that results in an increased secretion of glucocorticosteroids from adrenocortical cells [DOI: 10.1210/endo-125-5-2793, doi:10.1089/ars.2017.7186, doi:10.1055/s-2007-978915, doi:10.1210/edrv.21.1.0389, ...]. Numerous studies have shown that glucocorticoids play a key role in starvation-induced muscle atrophy via GR in myofibers [DOI: 10.1172/JCI38770, <https://doi.org/10.1152/ajpendo.00512.2011>, <https://doi.org/10.1016/j.biocel.2013.05.036>, doi: 10.3389/fphys.2015.00012, DOI: 10.1042/bj3070639 ...], in agreement with the data presented in Fig. S3. Moreover, when endogenous GC production in rodents is prevented by adrenalectomy, muscle atrophy induced by fasting is no longer observed (DOI: 10.1152/ajpendo.1993.264.4.E668).

Upon starvation, we show that the muscle weight of control mice is strongly decreased, that the expression of various well-known GR targets is induced in skeletal muscle, including *Atrogin1 (Fbxo32)*, *Murf1 (Trim63)* and *Redd1 (Ddit4)*, and that GR is bound to these genes by ChIP-seq, thereby showing that glucocorticoid signaling is active (Fig 3b, h and 4e).

In addition, dietary restriction reduces spleen mass (DOI: 10.1007/s11357-018-0022-2, DOI: 10.1242/jeb.153601), due to increased glucocorticoid levels that induce splenocyte apoptosis (DOI: 10.1677/joe.0.1630543). Spleen weights were taken as a measurement of equivalent glucocorticoid activity, and confirmed an active signaling with a more than 50% decrease in mass (Fig. 3b and S3d).

6. The authors have used tamoxifen-dependent CreERT2 system-generated LSD1(i)skm^{-/-} mice. Studies have shown that tamoxifen induces toxicity in muscles. How were the DEX injection experiments performed post-tamoxifen injection? The methods are unclear in the manuscript.

It was shown that long-term treatment of breast cancer with tamoxifen can induce myalgia, muscle aches and cramps in rare cases. No other signs of toxicity have been reported at recommended doses to our knowledge (doi:10.1007/s00280-014-2605-7).

In our model, mice are treated for 5 consecutive days with Tamoxifen to induce CreER^{T2} translocation to the nucleus, and DEX treatments were performed one month after TAM injections (page 18, line 426). Previous studies from our group showed that tamoxifen levels are neglectable already 3 days after the last injection since the CreER^{T2} expression was mainly cytoplasmatic at this time (<https://doi.org/10.1006/meth.2001.1159>). Note that control mice were also treated with TAM and do not present histological defects (new Fig. S3c), thus preventing a potential bias caused by this compound.

7. Muscle-specific and whole-body LSD1^{-/-} mice are the animal models used in this study. However, this mouse is not characterized well. It is essential to characterize the mouse model for various physiological parameters, including age-wise characterization of body weight, food, and water intake, blood profiling, etc. Also, characterize overall muscle phenotype in these mice under basal conditions by performing treadmill run experiments, metabolic cage experiments, and SDS COX staining at the least.

As mentioned in point 4, we did not use whole-body LSD1^{-/-} mice since it is lethal, but muscle-specific LSD1 knock-out mice. Since this manuscript focuses on the role of LSD1 in GC-induced muscle atrophy, we believe that characterizing the role of LSD1 in metabolism is out the scope of this study. Our data nevertheless show that muscle histology of mice with myofiber-specific LSD1 ablation does not lead to gross abnormalities (page 5 lines 116-119, Fig. S1k). In addition, body, muscle and WAT weight, as well as muscle strength are displayed in new Fig. S1l, m, and are similar between control and LSD1^{skm^{-/-}} mice. Muscle weight and strength, as well as transcript levels of well-known atrogenes (*Atrogin/Fbxo32* and *Murf1/Trim63*) were determined to validate muscle atrophy (new Fig. 5a-c, S5a-c and 7a-c).

In this work, the authors may need to validate the muscle atrophy by examining the muscle fiber type upon dexamethasone treatment.

It is well-known that elevation of GC levels leads mainly to type 2B myofiber atrophy (<https://doi.org/10.1016/j.cmet.2011.01.001>, DOI: 10.1152/jappl.1995.78.2.629, <http://dx.doi.org/10.1016/j.mce.2013.03.003>). In light to these data, we show that fast-twitch muscles such as gastrocnemius and quadriceps are the most affected by DEX treatment. We now provide evidence that soleus muscle weight is not altered by GCs (Fig. 5a, S5a and 7a). A sentence explaining the differences between slow- and fast-twitch muscles' DEX response has been added in the discussion (page 17 lines 399-402).

8. It is unclear whether LSD1 has a cell-autonomous role in skeletal muscle. The authors may perform in vitro experiments involving knock-down and overexpression of LSD1 using C2C12 or murine primary myotubes to rule out any possibility of paracrine and autocrine signaling influencing the muscle phenotypes in the mice models used for this study.

To demonstrate that LSD1's function in skeletal muscle is cell-autonomous, we engineered and analyzed two muscle-specific mouse models of LSD1 ablation. We demonstrate that LSD1 binding at DNA in muscle is abolished in LSD1^{(i)skm-/-} mice, thereby showing its myofiber specificity.

In addition, we used in the original version of the manuscript the LHCNM2 tissue culture model of human myofibers to test the GR-LSD1 functional interaction in the presence or absence of ligand (former Fig. 7, new Fig. 6). We now provide further evidence that GR/LSD1/NRF1 target gene expression is decreased upon silencing (new Fig. 2m and S2i). Thus, while we acknowledge that in vitro models offer more direct prove of cell-autonomous functions, we believe that our in vivo findings provide substantial evidence of the importance of LSD1 in skeletal muscle functionality. It also captures the complex nature of physiological responses, which often gets diluted in isolated cell cultures.

9. Numerous critical concerns exist with the immunoprecipitation experiments in the paper:

a. The authors have used 10 % input, and most of the immunoprecipitated lanes have stronger bands than the input lane, which is inappropriate and impossible.

For IP experiments, 10% of the protein extract were used for inputs, and the 90% left were immunoprecipitated. If the efficiency of the IP is of 100%, the band would be 9 times stronger than that of the input. In our case, the quantification reveals that we never exceed 3 times the intensity of the input. Please find here various articles with similar observations:

For LSD1: DOI: 10.1093/nar/gks031, 10.1038/s41388-021-02123-7, 10.7554/eLife.84618

For GR: DOI:10.4049/jimmunol.1201776, 10.3390/ijms24087130, 10.1093/nar/gkab226

b. Fig. 1B, 2D, 2F, 3A, 5F: It is unexpected to observe greater expression of the interacting partner protein than the immunoprecipitated protein.

We apologize for this lack of precision. From 200 ug of protein extracts IPed with LSD1 antibody 30 ul of proteins were loaded for GR detection and 10 ul for LSD1 detection, and vice versa for GR IP. This allows to avoid that the signal from the IP masks that of the co-IP. This procedure has been used for the various panels. This point is now stated in the method section (page 27, lines 630-633).

c. Also, the authors have used single IgG control for two independent immunoprecipitation reactions performed with two different antibodies, which may also be wrong. Please confirm if the origin of the secondary antibodies against LSD1 and GR primary antibodies is the same. Please verify the results and add loading controls to the panels.

Rabbit IgG has been used as been used as control since NRF1, LSD1 and GR antibodies were generated in this species (page 27, lines 629). In general, loading controls are not presented for IP experiments as they are not informative (please see abovementioned articles among others).

d. Note that, oddly, the input bands in the top and bottom panels are non-identical in Fig. 1B.

As mentioned in point 9b, for two-directions IP experiments, the IPed material was split in two and loaded on two different gels since the molecular weight of GR and LSD1 is too close, and striping the membrane might result in signal lowering.

e. In Fig. 4G: Are the IgG antibodies used for Trim63 and Ddit4 of different origins?

There was no Fig. 4g in the original version of the manuscript. However, in Fig. 5g, ChIP-qPCR was performed with anti-GR and anti-LSD1 antibodies produced in rabbit. Thus, IgGs from that species were used for the analysis of the two loci presented.

10. Critical controls are missing in the figures. In Figure 6, (A-E), extended figures 6 and 7B- the CC-90011 alone treatment group is missing. Without this group, it would be hard to interpret the data shown in Figures 6, extended Figures 6 and 7.

Our primary aim was to determine the implications of LSD1 modulation on DEX-induced muscle atrophy. The design of our experiments and controls was intended to distinguish the combined effects of the inhibitor and DEX. To determine the independent effects of the LSD1 inhibitor on GR activity, CC-90011 control has now been included in the various panels of new Fig.6, 7, S7.

11. Fig. 1G-I: Given that GR is a major transcription factor for atrogene, it is crucial to reveal whether LSD1 interacts with the atrogene loci in myofibres at physiological GC levels.

To address this point, two examples, namely *Fbxo32* (*Atrogin1*) and *Trim63* (*Murf1*) loci, were provided in Fig. S1f in the original version of the manuscript. We now provide additional examples in new Fig. and S2g.

12. Fig. 6A, B: It is difficult to interpret the effect of LSD on DEX-induced muscle atrophy without first ascertaining the impact of the LSD1 inhibitor itself on GR activity. It is strongly recommended to validate the inhibitor in the mouse model used.

We appreciate the constructive feedback on the interpretation of the effects of LSD1 on DEX-induced muscle atrophy in original Fig. 6a and 6b (new Fig. 7a, b), and implemented the figure as mentioned in point 10. Of note, the effect of CC-90011 on GR activity was assessed by ChIP-qPCR as CC-90011 decreases GR binding to DNA (new Fig. 7c).

13. Since the primary outcome of the paper is to characterize the use of LSD1 inhibitor to circumvent the muscle atrophy phenotype associated with the use of DEXA as an anti-inflammatory drug, it is crucial to observe the cytokine and chemokine profile in DEX-treated control and LSD1^{-/-} mice, in addition to the immune cell population shown in the manuscript.

We agree with Reviewer 1 on the importance of the impact of the LSD1 inhibitor on DEX-associated muscle atrophy in parallel with GR anti-inflammatory activities. To this aim, we provide two additional lines of evidence.

Using naive Th0 lymphocytes or Th17 differentiated cells, we now show that whereas a 24h DEX treatment at 100 nM decreases IL17 production by more than 3 folds, a 24h CC-90011 administration at 100 nM had no effect on interleukin production, and did not impact DEX-dependent decrease in IL17 levels (new Fig. 7f).

Moreover, the analysis of an experimental model of inflammatory bowel disease shows that LSD1 inhibition does not affect the anti-inflammatory effects of DEX on colon, but protects fast-twitch muscles from atrophy (new Fig. 8).

14. Fig. S1H-I: Please assess the extent and tissue specificity of LSD1 depletion in the mouse model by western blotting. The authors may test the expression of LSD1 in different muscles and organs.

We now added the western blot characterization of LSD1 protein levels in gastrocnemius, soleus, tibialis and quadriceps muscles, as well as in epididymal white adipose tissue and spleen (new Fig. S1i and S3a). Moreover, immunofluorescent detection of LSD1 clearly shows that LSD1 is expressed in myofiber nuclei of control mice but not in those of LSD1^{skm^{-/-}} and LSD1^{(i)skm^{-/-}} mice (Fig. S1j and S3b).

15. Fig. 3: To assess the role of the GR/LSD1 complex in muscle atrophy, it is imperative to evaluate the activity of GR under starvation conditions in control and LSD1^{skm^{-/-}} mice.

As mentioned in point 5, GR activity is stimulated under starvation conditions in WT mice, as shown by increased transcript levels of classical GR target genes such as *Atrogin1*, *Murf1* and *Ddit4* (Fig. 3h and

S3I), associated with GR binding at these loci (Fig. 4e). Moreover, the data show that the phenotype of $LSD1^{(i)skm-/-}$ mice parallels that of $GR^{(i)skm-/-}$ mice in our experimental settings. Notably, the atrogenes are much less induced by starvation in both $LSD1^{(i)skm-/-}$ and $GR^{(i)skm-/-}$ mice. This observation strengthens our hypothesis regarding the interplay between LSD1 and GR in regulating muscle response to starvation.

16. Fig. S4F: Given that FAD is depletion upon starvation, the enzymatic activity of LSD1 may be expected to reduce upon food deprivation. It is, therefore, important to investigate and comment upon the time-sensitive role of LSD1 in GR-mediated activation of starvation-induced muscle atrophy.

We agree with the referee and implemented this point in the discussion, paralleling the data obtained in Fig. S4f with the ChIP-qPCR kinetics shown in Fig. 4e. This point was discussed in the original version of the discussion of the manuscript (page 14 lines 325-331, now page 19 lines 438-443).

17. Fig. S1J, 3A, 3E, 4C: Throughout the paper, authors have characterized LSD function almost exclusively in the gastrocnemius muscle. Verifying these critical results in other important muscle types, especially the tibialis anterior (among other Type I and Type II muscle fibers), would be interesting.

We used gastrocnemius as it is a glucocorticoid-sensitive fast-twitch muscle composed of various type-2 fibers. Nonetheless, our data provide evidence that all the analyzed fast muscles (gastrocnemius, tibialis and quadriceps) are similarly affected by either DEX or starvation, as shown by decreased muscle mass, contrary to soleus muscle (Fig. 3b-d, S3d, f, g, 5a and S5a).

As further evidence, we now provide western blot analysis of anabolic and catabolic pathway activities in quadriceps muscle (Fig. 3f, g, S3j-l, 5d, e and S5d-f). In addition, GR and LSD1 co-localization is now presented in the various muscles (new Fig. S1b).

18. Fig. 3E: Electron microscopy images are required to be quantified. Also, these results need to be verified by other methods.

The measure of muscle weight and CSA, as well as RT-PCR analysis on genes from the proteasome system and western blot analysis of actors of the protein degradation pathway provide quantitative datasets for muscle atrophy. We presented electron microscopy pictures to characterize myofibers' defects. Moreover, we mentioned in the original version of the manuscript that "ultrastructural analysis of gastrocnemius muscles revealed disruptions of myofibrils in more than 60% of the sarcomeres after starvation of control mice, with loss of myofilaments, rupture of Z-lines and enlarged sarcoplasm, whereas less than 20% of the sarcomeres were damaged in muscles of $LSD1^{(i)skm-/-}$ and $GR^{(i)skm-/-}$ mice". Note however that electron microscopy is a semi-quantitative method and should be considered as additional evidence for the phenotype.

19. Fig. 5D, E: Inconsistent use of loading controls in western blotting.

To be consistent, GAPDH western blot has been replaced by that of Tubulin in Fig. 5d (new Fig. 5f). Both are conserved in the source data file.

20. Fig. 4C, 5D-E: The authors have not quantified the western blots in this manuscript. All western blots need to be quantified and presented.

Protein quantification was already provided in Fig. 4d for Fig. 4c. Quantification has now been added next to the additional western blot panels.

21. Extended figure 3C- fiber cross-section area (CSA) of $GR^{(i)skm-/-}$ mice starved for 48 h is much higher than the $GR^{(i)skm-/-}$ mice fed animals, which I feel is impossible. Fig. S3C: This figure indicates that starvation induces hypertrophy in the muscles of $GR^{-/-}$ mice. This is highly improbable since starvation is well known to inhibit protein synthesis. Furthermore, Fig. S3H indicates upregulation in mRNA expression of atrogenes, contradicting the hypertrophy phenotype reported.

Original Fig. S3d (new Fig. S3g) shows no statistical difference in the average CSA of GR^{(i)skm^{-/-}} fed and starved mice. Moreover, a 2-way ANOVA analysis has been added on new Fig. 3c and S3f to show that there is no statistical difference in CSA distribution between these two conditions, even for fibers with an area >5000. The table recapitulating ANOVA's results is provided in the source data file.

Minor comments:

1. Please ensure uniform color coding across data figures in the manuscript.

This point has been implemented.

2. Fig. 1H, S1E: Since the study is focused on skeletal muscles, please avoid displaying cardiac-associated genes in the list to avoid confusion.

In the various figures showing pathways analysis, we presented the various bioinformatics annotations obtained from the KEGG analysis. We did not select any in particular. Note that the one entitled "MicroRNAs in Cardiomyocyte Hypertrophy" contains genes expressed in skeletal muscle.

3. Fig. 6, 7B: Please denote groups as Vehicle, DEX, and DEX+CC in the figure for simplicity.

This point has been implemented.

Reviewer #2 (Remarks to the Author):

In this manuscript, Cai and colleagues investigate the molecular details underlying glucocorticoid-induced skeletal muscle atrophy, which limits the use of synthetic GC in patients that require chronic administrations. They report that LSD1 interacts with both GR and NRF1 to regulate gene expression. Pharmacological inhibitors of LSD1 attenuate muscle atrophy, providing evidence for the involvement of LSD1 and therapeutic promising applications. The number of reports that establish a role for LSD1 in skeletal muscle is increasing, so is the tight relationship with steroid hormone receptors. The manuscript is very well written, results are clear and conclusions are appropriate. Before publication, we have the following suggestions that may improve to strength the authors' conclusions.

We thank the reviewer for his/her very positive feedback on our manuscript. His/her constructive suggestions will further improve the quality of our work.

1) In Fig. 1 the authors show colocalization of LSD1 and GR but not all the nuclei are positive, can the author stain for the different fiber types with MyHC marker subtype to establish whether there is a preferential expression of type IIb vs intermediate or oxidative fibers? Is there a homogeneous distribution of double positive nuclei in the entire transversal section, as the fiber type composition of gastrocnemius is not uniform.

We performed a co-staining with GR (mouse IgG1) /LSD1 (rabbit) in slow-, fast- and mixed muscles, but could not co-stain with MyHC as all these antibodies were produced in mouse. Our data show that GR and LSD1 evenly colocalize in the various analyzed muscles. Note that slow-twitch fibers are usually auto-fluoresce and are thus visible in green on our images (new Fig. S1b). Thus, even though GR and LSD1 co-localize in the different types of fibers, their action on muscle atrophy is specific for the fast-twitch ones, indicating that additional factors are selectively involved in fast/mixed-twitched fibers to induce atrophy. This point is now mentioned in the discussion (page 17 lines 399-402).

2) In the IP experiment, the lower band of LSD1, that also the authors notice, is interesting, could that be another isoform (alternative splicing) or another member of the KDM family that interacts with GR? IP with an anti-GR Ab pulls down a lower band that is not visible in the input, may be because it is not enriched enough, but the upper isoform is not visible. This is problematic. However, CHIP analysis shows interaction at the chromatin level. But the authors should clarify why the bands have different size, or reconsider the interpretation of the IP experiments. LSD1 has isoforms (2a, 8a, etc).

We appreciate the opportunity to clarify this aspect of our study. LSD1 has a molecular weight of around 93 kDa and usually migrates between 110 and 120 kDa. In skeletal muscle, two isoforms co-exist, the nLSD1 and the LSD1 2a that migrates 5 kDa upper than nLSD1 (Fig. R3A). The presence of these two isoforms was confirmed by Sashimi plot analysis obtained from RNA-seq datasets of gastrocnemius muscle (Fig. R3B). Note that the two bands are not always visible according to the percentage of the gel or the mount of protein loaded.

Upon immunoprecipitation with GR or NRF1 antibodies, and LSD1 to a less extent, LSD1 was detected at a molecular weight between 100 and 110 kDa, which was shown by using both N- and C-terminal LSD1 antibodies (Fig. R3C). Since similar data were obtained with anti-GR and anti-NRF1 antibodies, we did not question the antibody use for IP. We then questioned whether this issue was a problem of species since LSD1 antibodies were originally designed for human usage. Thus, we tested LSD1/GR interaction in LHCNM2 cells, but evenly obtained the band between 100 and 110 kDa upon GR IP (Fig. 6b).

After unsuccessfully testing various post-translational modifications, we thus suspected a technical issue. After testing various parameters, we found that the observed molecular weight discrepancy originates from LSD1 sensitivity to salt and/or pH composition of the elution buffer.

In summary, the samples obtained from IP contain concentration of salt, ion chelating agents and pH values that differ from the input samples, and can affect the migration rate of the LSD1 protein within the acrylamide gel. Adjusting this parameter by adding for instance RIPA buffer to the IP samples mitigates this issue (Fig. R3D).

Figure R3: A. Western blot analysis of LSD1 levels in gastrocnemius muscles of wild-type mice on a 8% gel. B. Sashimi plot for alternatively spliced exon and flanking exons in four wild-type samples. Per-base expression is plotted on y-axis of Sashimi plot, genomic coordinates on x-axis, and mRNA isoforms quantified are shown on bottom (exons in blue, introns as lines with arrow heads). nLSD1 and LSD1 2a isoforms are shown for each sample. C. Representative western blot analysis of gastrocnemius muscle nuclear extracts from wild-type mice immunoprecipitated with anti-GR or anti-LSD1 antibodies. Membranes were decorated with N-terminal (3544) and C-terminal (20752) anti-LSD1 antibodies. IgG served as a control for immunoprecipitation. D. Representative western blot analysis of gastrocnemius muscle nuclear extracts from control (Ctrl) and LSD1^{skm}^{-/-} mice immunoprecipitated with anti-GR or anti-LSD1 antibodies and eluted in the presence of lysis buffer. Membranes were decorated with anti-LSD1 antibodies. IgG served as a control for immunoprecipitation.

3) Were the CHIP on target validations performed on different samples as those used for CHIPseq? If so, this strengthens the validation analysis.

ChIP-seq and ChIP-qPCR analyses were performed on independent cohorts. This point has been implemented in the method section of the revised manuscript (page 28 line 674).

4) The first sentence of the figure captions may be more informative with respect to “Characterization of...”, “Role of...” can it be a sentence summarizing the main message of the figure?

As proposed by the referee, the captions have been modified.

5) Is the interaction of GR with LSD1 and NRF1 modulated by ligand? Can this be tested in cells treated with vehicle or GC?

New Fig. 5g now provides evidence that the interaction of GR with LSD1 and NRF1 is not affected modulated by pharmacological levels of DEX. However, circulating glucocorticoids are present in endogenous conditions. Thus, in addition to these data, we performed coIP experiments in LHCNM2 cells grown for 24h with charcoal-treated or in full medium (Fig. R4 and new Fig. 6b). The decrease in the interaction in charcoal-treated cells indicates that the interaction is ligand-dependent.

Figure R4: Representative western blot analysis of nuclear or cytoplasmic extracts of LHCN-M2 cells grown in full-medium or charcoal-treated medium, immunoprecipitated with anti-GR or anti-LSD1 antibodies. Rabbit IgG served as a control for immunoprecipitation.

6) The observation that LSD1 is required for interaction of GR with NRF1 (Fig 2F) suggests a specific effect, is it possible that LSD1 demethylates GR? Or is this effect (fig 2G) occurring at the chromatin level?

To thoroughly address the question of LSD1 demethylase activity on GR, we will combine this answer with that of point 9 (see hereafter).

7) I suggest to move to main figure 3 analysis of CSA in control and starved mice with the relative quantification. Relative to autophagy, WB analysis of p62 and LC3 I/II should be added to conclude that autophagy is not induced and to determine if the autophagy flux is regular.

As proposed by the referee, the analysis of CSA in control and starved mice has been moved to main figure 3 and we performed western blot analysis of p62 and LC3 I/II, the expression of which is similar between control and LSD1 mutant mice (Fig. 3f, g and 5d, e).

8) What would be the benefit to use CC-90011 with respect to TCP, for instance?

Tranylcypromine is a cell-permeable phenylcyclopropylamine that inhibits the monoamine oxidase and histone demethylase activities, respectively, of MAO A/B ($K_i = 101.9$ and 16.0 M, respectively) and LSD1/2 ($K_i = 242.7$ and 180.0 M, respectively), four members of a flavin-dependent amine oxidase family enzymes, by a covalent adduct formation with the enzyme-bound FAD, and is therefore not specific for LSD1. Tranylcypromine irreversibly inhibits LSD1 with an IC_{50} value of $20.7 \mu\text{M}$.

Pulrodestat (CC-90011) is a potent, selective, reversible and orally active inhibitor of lysine specific demethylase-1 (LSD1) with an IC_{50} of 0.25 nM, which justified our choice. A sentence clarifying this point has been added in the discussion (page 19 lines 444-449).

9) The effect of the inhibitor (occurring through the catalytic activity) implies that LSD1 demethylase activity is important, is it only involved in histone modification or is it the result of direct modification of GR or NRF1?

LSD1 has been shown to demethylate the non-histone protein P53 (DOI: 10.1038/nature06092). To our knowledge, there has been no study describing NRF1 and GR methylation (see Gene cards website and the following reviews: DOI: 10.1016/j.biopha.2023.115145 and 10.1074/jbc.272.30.18732). Therefore, we tested if these two factors are methylated. Using mouse muscles, we performed IP against GR and NRF1, followed by WB with pan-Methyl-Lysin antibody. Our results did not reveal GR or NRF1 methylation in muscle extracts (Fig. R5), thereby strongly indicating that LSD1 cannot further demethylate the two proteins.

Figure R5: Pan-methyl-Lysin western blot analysis of mouse muscle cell extracts immunoprecipitated with anti-GR, anti-NRF1 or rIgG antibodies.

Thus, we speculate that in our context LSD1 holds two main functions: the first one is to demethylate H3K9me1/2 to promote gene transcription, and the second one is a scaffolding function to allow GR and NRF1 interaction.

10) And what is the involvement of NRF1 in muscle atrophy? What do the data shown in Fig 2 add to the rest of the story?

NRF1 has previously been proposed to be a pioneer factor based on its ability to bind *de novo* hypomethylated DNA sites (<https://doi.org/10.1038/nbt.2798>). NRF1 only bears canonical hallmarks of a pioneer factor in the absence of DNA methylation, where it can bind autonomously. These genome-wide analyses thereby revealed that NRF1 occupies several thousand additional sites in the unmethylated genome, resulting in increased transcription. In a previous study, we showed that GR at skeletal muscle-specific enhancers interacts with NRF1 located at open-chromatin promoter regions of target genes via the formation of chromatin loops, to stimulate gene transcription. Fig 2. of the current manuscript shows the molecular mechanism by which LSD1 mediates glucocorticoids' action by allowing interaction between GR at enhancers and NRF1 at promoters at physiological GC levels (see

also Reviewer #1 points 2 and 8 and Reviewer #2 point 5), via a direct interaction with the two transcription factors. Pathway analysis revealed that the GR-LSD1-NRF1 complex is mainly bound to genes involved in protein degradation. We show in Fig. 5g that LSD1 is still in complex with GR and NRF1 to promote target gene expression such as *Murf1* and *Trim63*. Thus, NRF1 is not a regulator *per se* of muscle atrophy but is required for GR-dependent gene expression via LSD1 (new Fig. 2m and S2i).

11) The discussion and the manuscript would benefit if the authors take into consideration recent findings that establish a role of LSD1 in muscle in the context of another steroid receptor, androgen receptor. Especially considering the anabolic effects of androgen signaling in muscle. LSD1 represses transcription of several genes, but it transactivates AR, is this the case for GR? Similar specific effect that boosts GR transactivation?

Our data obtained from CHIP and RT-qPCR strongly indicate that LSD1 is required for GR-dependent transcription. Similar to what has been shown for AR, luciferase assay at GR response element shows that LSD1 overexpression enhances GR activity in the presence of DEX (Fig. R6), confirming that LSD1 acts as a co-activator of GR. The discussion has been implemented accordingly (page 14 lines 372-376).

Figure R6: Transactivation analysis in LHCNM2 myofibers transfected with a pGL3-promoter vector containing the 5'-GGAACAGAACACGGTGTAGCTGGGA-3' GR response element, 100 ng of a pSG5-GR expression vector, and 37.5 ng of a pSG5-LSD1 expression vector, in the presence or absence of 10 nM DEX. Data were normalized to Firefly expression.

12) For statistical analysis, ANOVA (not t test) shall be used for more than 2 sample comparisons.

T-test has been used for statistical analysis of Fig. 1f, in which each antibody was compared to its respective IgG control. Since we did not aim comparing LSD1 with GR antibody, we did not perform ANOVA test. Fig. 2g has been reanalyzed with a One-way ANOVA with Tukey correction.

Reviewer #3 (Remarks to the Author):

The study by Cai and collaborators investigates the relationship between the lysine-specific demethylase LSD1 and the glucocorticoid receptor (GR) in the context of muscle atrophy. The Authors show that these molecules directly interact resulting in the modulation of target gene expression. Among others, gene involved in the regulation of muscle mass homeostasis are targeted by LSD1 and GR interaction. The study is mainly based on genetic approaches, also taking advantage of mice in which the expression of GR or LSD1 is down-regulated. The results showing the interaction between LSD1 and GR are quite convincing, while few notes of care arise as for the relevance of such interaction to muscle atrophy, which is investigated in mice exposed to fasting or administered dexamethasone (DEX). Specifically, the following issues should be taken into consideration:

We thank the reviewer for his/her constructive comments. We have implemented the suggested modifications.

1) line 85-86: while it is conceivable that some GRs are located in the nucleus also in physiological conditions such as those analyzed by the Authors, Figure 1A shows no or very little cytoplasmic GR, while this latter should be its main location in this experimental setting;

Regarding GR localization in Fig. 1a, it is important to note that, even in physiological conditions, circulating GC levels can induce GR translocation to the nucleus as shown previously (DOI: 10.1093/nar/gkab226). Please find additional GR staining in mouse prostate in which GR is also essentially nuclear (Fig. R7).

Figure R7: Immunofluorescent detection of GR (red) in mouse prostate and skeletal muscle of 10-week-old mice. Nuclei are stained with DAPI (blue).

2) one of the main findings reported in the present study is that muscle-specific lack of LSD1 confers a significant resistance to wasting induced in mice (age? initial body weight?) by 48 h fasting. Such a pattern is associated with inhibition of the expression of genes involved in catabolic pathways such as muscle-specific ubiquitin ligases and autophagy and with preservation of muscle ultrastructure. However, these results do not rule out that other pathways in addition to the GR-dependent one(s) could be altered by the lack of LSD1 in the skeletal muscle. In this regard, 48 h fasting results in a severe substrate shortage. While adipose tissue reduction, more marked in LSD1^{-/-} mice than in controls, likely

attempts to compensate such a shortage, it is hardly able to achieve a total 'buffer' against the lack of nutrients; it would have been interesting to see the effects of LSD1 inhibitor in fasted animals;

We thank the reviewer for the interesting comment. As requested, the initial body weight of the mice used for the starvation experiment is now shown in Fig. 3b and S3d. In addition, the age at which starvation was performed is now stated in the Methods section (page 21 line 489) and in the figure legends.

As mentioned in the manuscript, starvation is not a good model to treat with an LSD1 inhibitor as LSD1 is released from the chromatin 6h after food deprivation. Moreover, such an experiment has little clinical relevance.

3) the second main observation of the study is that LSD1 down-regulation protects mice against DEX-induced atrophy, apparently without affecting DEX anti-inflammatory properties. However, only indirect evidence is provided in this regard, showing that spleen changes in size and both spleen resident and circulating immune cell sub-populations do not differ in the presence or in the absence of the LSD1 inhibitor. In this regard, the lack of effect on the anti-inflammatory properties should be demonstrated using the LSD1 inhibitor on models such as the experimental autoimmune encephalomyelitis or inflammatory bowel disease;

As proposed by the referee, we used an experimental model of inflammatory bowel disease. Our data provide evidence that LSD1 inhibition under these experimental conditions also does not affect the anti-inflammatory effects of DEX, and partially protects fast-twitch muscle from atrophy (new Fig. 8, and result section pages 14-15).

4) control mice treated with the LSD1 inhibitor are not included in the experimental design, while they should;

As proposed by the referee, the CC-90011 control has been included in new Fig. 6, S6 and 7.

5) LSD1 lack of action on the anti-inflammatory activity does not really fit with data reported in Figure 1I, which shows an up-regulation of genes involved in the Inflammatory Response Pathway;

It is correct that Fig. 1i shows that LSD1 and GR are involved in the transcriptional regulation of genes involved in the Inflammatory Response Pathway in skeletal muscle. However, this does not imply that they will carry out a similar function in immune cells or in spleen. Indeed, whereas GR transcriptional activity is mainly driven by its response element (GRE) in myofibers, it is thought to act via the transrepression of transcription factors such as NFkB, AP1 or STAT3 to control the immune response. A sentence to discuss this point has been added in the discussion (page 19 lines 457-461).

6) most of the results obtained in starved or DEX-exposed mice refer to gene expression data. However, since changes of mRNA levels do not necessarily match with protein expression and activity, at least some of the markers of protein hypercatabolism should be assessed at the protein level;

As proposed by the referee, the western blot analysis of genes involved in the AKT/mTOR/FOXO pathway has been included, confirming that GR^{(i)skm^{-/-}} and LSD1^{(i)skm^{-/-}} mice are resistant to fasting- and DEX-induced protein catabolism (Fig. 3f, g, S3i-k, 5d, e and S5d-f).

7) data on human myotubes do not add any particular cue, they could be cited as supplemental;

We used human myotubes as a confirmation of LSD1 cell-autonomous effects. We now provide additional evidence on the formation of the GR/LSD1 complex in new Fig. 6b. We implemented this dataset with primary Th0 and Th17 cells to modelize the effects of DEX in combination with CC-90011 on inflammation (new Fig. 7f).

8) it is not correct to talk about 'muscle catabolism' or 'muscle degradation'. Indeed, a tissue per se cannot be degraded or synthesized, rather its components (protein, lipids, etc.) are processed during anabolic/catabolic reactions.

These points have been implemented. “Muscle catabolism” has been modified by muscle wasting (Page 2 line 32), muscle atrophy (Page 4 line 74), muscle wasting (Page 17 line 449), and “muscle degradation” by muscle wasting (Page 10 line 234).

Similarly, it is not correct to talk about muscle atrophy when referring to an in vitro system (line 56); We modified “muscle atrophy” by “protein degradation” (Page 12 line 299).

9) finally, in lines 203-204 LSD1 is reported to play a role in ‘initiating the food deprivation process’, which is a non-sense, since such a process starts when mice are no more allowed to reach food. We apologize for this lack of precision and modified the sentence now page 10 line 246 in the new version of the manuscript.

REVIEWER COMMENTS

Reviewer #1 (Remarks to the Author):

The authors have adequately addressed my concerns and the quality of the manuscript has increased significantly.

Reviewer #2 (Remarks to the Author):

The authors have addressed all my concerns, the revisions are solid and we have no further comments.

Reviewer #3 (Remarks to the Author):

The present version of the study by Cai and collaborators is substantially improved. Most of the criticisms have been satisfactorily addressed, although few points still require attention:

in their reply, the authors do not clarify about the possibility that pathways other than the GR-dependent one can be involved in the resistance to fasting-induced wasting reported in mice featuring muscle-specific lack of LSD1;

the authors seem to have merged two different comments, the one about additional pathways (see previous point) and the one asking for information about LSD1 inhibition in fasted animals. With reference to this latter issue, the authors say that this experiment would not have clinical relevance. First of all, being the study not a clinical one, this statement does not seem appropriate. This consideration apart, the reviewer thinks that understanding if the effects obtained by tissue-specific gene k.o. can be recapitulated by a sistemically administered drug is of great translational interest. In this regard, the authors argument that since 'LSD1 is released from the chromatin 6h after food deprivation', fasting is not an adequate model for studying the use of LSD1 inhibitors. This issue could be circumvented by pre-treating the animals with the inhibitor;

with reference to answer n. 6, I would remind the authors that changes in molecular markers, even at the protein levels (available in the present version of the paper), are just suggestive of protein catabolism (or synthesis) changes, which are not directly assessed. Along this line I encourage the authors to search and correct the text accordingly. This includes the change introduced at page 12, line 299: the authors did not measure protein degradation in DEX-treated myotube cultures. The word 'atrophy' referred to myotubes could be replaced by 'thinning', or 'reduction in size'.

Reviewer #1 (Remarks to the Author):

The authors have adequately addressed my concerns and the quality of the manuscript has increased significantly.

We are pleased that our revisions have satisfactorily addressed the reviewer's concerns.

Reviewer #2 (Remarks to the Author):

The authors have addressed all my concerns, the revisions are solid and we have no further comments.

We thank the reviewer for acknowledging the improvements made to our manuscript.

Reviewer #3 (Remarks to the Author):

The present version of the study by Cai and collaborators is substantially improved. Most of the criticisms have been satisfactorily addressed, although few points still require attention:

We thank the reviewer for his/her positive feedback on our manuscript.

1. In their reply, the authors do not clarify about the possibility that pathways other than the GR-dependent one can be involved in the resistance to fasting-induced wasting reported in mice featuring muscle-specific lack of LSD1;

This point has now been clarified in the discussion (pages 17-18, lines 406-417).

2. The authors seem to have merged two different comments, the one about additional pathways (see previous point) and the one asking for information about LSD1 inhibition in fasted animals. With reference to this latter issue, the authors say that this experiment would not have clinical relevance. First of all, being the study not a clinical one, this statement does not seem appropriate. This consideration apart, the reviewer thinks that understanding if the effects obtained by tissue-specific gene k.o. can be recapitulated by a systemically administered drug is of great translational interest. In this regard, the authors' argument that since 'LSD1 is released from the chromatin 6h after food deprivation', fasting is not an adequate model for studying the use of LSD1 inhibitors. This issue could be circumvented by pre-treating the animals with the inhibitor;

Following the suggestion from the reviewer, we have updated our manuscript to underscore the potency of the LSD1 inhibitor in a clinically relevant mouse model for experimental colitis in the first round of revision. We have thereby demonstrated that LSD1 inhibition effectively prevents muscle wasting associated with dexamethasone treatment, while preserving its anti-inflammatory activities (see Figure 8).

The reviewer has expressed interest in knowing whether the LSD1 inhibitor can also counteract fasting-induced muscle wasting as seen in LSD1^{(i)skm-/-} mice, highlighting its considerable translational potential. To address this question, the reviewer suggests pre-treating the animals with the inhibitor to explore this possibility. However, the compound CC-90011 is characterized by a reversible action with a brief half-life of merely two hours (<https://www.medchemexpress.com/pulrodemstat.html>, doi: 10.1021/acs.jmedchem.0c00978, DOI: 10.1002/cncr.34366), leading to its partial degradation before the onset of catabolic pathways and subsequent replacement by FAD. This situation presents substantial challenges for data interpretation, given the intricate dynamics involved. Therefore, addressing this issue would require extensive optimization of several aspects, such as the mode of administration and the appropriate dosages of the compound. These requirements would complicate the comparison with other datasets presented in the manuscript and necessitate large cohorts of mice subjected to starvation, thus eliciting significant ethical concerns. After careful consideration of the benefits versus the ethical implications, the study's contributors have concluded that the prospective advantages of this investigation do not adequately outweigh the ethical dilemmas it presents.

Considering that the primary aim of our manuscript is to explore the effects of the LSD1 inhibitor on the beneficial and adverse outcomes of synthetic glucocorticoid use, we believe that the complex issue raised by the reviewer is beyond the scope of our current study.

3. With reference to answer n. 6, I would remind the authors that changes in molecular markers, even at the protein levels (available in the present version of the paper), are just suggestive of protein catabolism (or synthesis) changes, which are not directly assessed. Along this line I encourage the authors to search and correct the text accordingly. This includes the change introduced at page 12, line 299: the authors did not measure protein degradation in DEX-treated myotube cultures. The word 'atrophy' referred to myotubes could be replaced by 'thinning', or 'reduction in size'.

We have revised the manuscript using more precise and indicative language to describe the underlying biological processes, to better align with the indirect nature of the evidence. Those changes have been implemented throughout the new version of the manuscript (see modifications in green, e.g., page 10, lines 226-228, page 12, lines 279-281).

REVIEWERS' COMMENTS

Reviewer #3 (Remarks to the Author):

The authors have addressed the concerns raised during the revision, no further changes are required.